# Energetic and Structural Properties of Two-Dimensional Trapped Mesoscopic Fermi Gases

**Emma K. Laird[1,★], Brendan C. Mulkerin[2], Jia Wang[3], and Matthew J. Davis[1]**

**1** ARC Centre of Excellence in Future Low-Energy Electronics and Technologies, University of Queensland, Saint Lucia Queensland 4072, Australia
**2** ARC Centre of Excellence in Future Low-Energy Electronics and Technologies, Monash University, Clayton Victoria 3800, Australia
**3** Centre for Quantum Technology and Theory, Swinburne University of Technology, Hawthorn Victoria 3122, Australia

★ e.laird@uq.edu.au

## Abstract

We theoretically investigate equal-mass spin-balanced two-component Fermi gases in which pairs of atoms with opposite spins interact via a short-range isotropic model potential. We probe the distinction between two-dimensional and quasi-two-dimensional harmonic confinement by tuning the effective range parameter within two-dimensional scattering theory. Our approach, which yields numerically exact energetic and structural properties, combines a correlated Gaussian basis-set expansion with the stochastic variational method. For systems containing up to six particles, we: 1) Present the ground- and excited-state energy spectra; 2) Study non-local correlations by analysing the one- and two-body density matrices, extracting from these the occupation numbers of natural orbitals, the momentum distributions of atoms and pairs, and the molecular 'condensate fraction'; 3) Study local correlations by computing the radial and pair distribution functions. This paper extends current theoretical knowledge on the properties of trapped few-fermion systems as realised in state-of-the-art cold-atom experiments.

# 1   Introduction

Many-body quantum systems are generally intractable due to their vast complexity and numerous degrees of freedom. A few of the simplest cases — such as the Lieb–Liniger model of the one-dimensional Bose gas or the one-dimensional Fermi–Hubbard model — admit exact analytical solutions because they are integrable, but these are rare exceptions. One promising strategy for discerning how many-body features emerge in more realistic settings is to probe the underlying physics from the few-body limit. Since the two-body system is typically well characterised, a 'bottom-up' approach can be employed in which the number of particles is increased one by one, thereby introducing complexity in a controlled and stepwise manner. In this way, mesoscopic observables are often found to converge surprisingly rapidly toward the predictions of many-body theories, once those predictions are rescaled to account for varying particle number [1–9].

An experimental bottom-up approach has been realised by the research group of Selim Jochim using a tightly focused optical microtrap ('optical tweezer'). By superimposing this microtrap onto a large reservoir of ultracold fermionic atoms and gradually lowering its depth, a chosen small number of particles can be deterministically prepared in the ground state of a harmonic oscillator potential at temperatures close to zero [1, 8–10]. Applying this method to two-component Fermi gases, their experiments have shown that in quasi-one-dimensional geometries a many-body Fermi sea can form from only four atoms [1]. In quasi-two dimensions many-body 'Cooper-like' pairing — evidenced by a peak in the correlations between particles with opposing spins and momenta at the Fermi surface — has been experimentally observed with as few as twelve atoms [9].

To better understand the latter experiment, in Ref. [11] we theoretically modelled an increasing number of spin-balanced two-component fermions confined in a quasi-two-dimensional harmonic trap. Our numerical approach — commonly referred to as the explicitly correlated Gaussian (ECG) method [12–15] — combined a stochastic variational framework with the use of ECG basis functions [16, 17], allowing us to compute experimentally measurable observables with very high accuracy. In particular, we calculated the lowest monopole excitation energies and ground-state opposite-spin pair correlations as functions of increasing attractive interaction strength [11]. The few-body physics was captured by applying two-dimensional scattering theory [18–20] to a finite-range Gaussian interaction potential, with the effective range tuned to model realistic quasi-two-dimensional scattering [21–24]. For gases comprising up to six equal-mass fermions, we found that time-reversed pairing in the ground state was predominant at momenta significantly below the Fermi momentum [11]. Together with experimental findings [9], this suggested that the Fermi sea — which, beneath the Fermi surface, Pauli-blocks the superposition of momenta required to form a paired state — must emerge in the transition from six to twelve particles.

Here, we apply the ECG method to the same Fermi gases to obtain new energy spectra and ground-state structural properties, which are crucial for their theoretical characterisation and

thereby further advance our understanding of fermionic few-body systems. This paper is organised as follows: In Section 2 we outline our model and the underlying two-body scattering theory. Section 3 details our results: In Subsection 3.1 we generate the energy spectra of the ground state and low-lying excited-state manifolds for gases containing two, four, and six particles. We quantify non-local correlations between the trapped fermions by analysing the one- and two-body density matrices in Subsection 3.2. In Subsection 3.3 we analytically Fourier transform the density matrices to extract the momentum distributions of individual atoms and opposite-spin pairs. To quantify local correlations in the Fermi gases we examine the radial and pair distribution functions in Subsection 3.4. In Subsection 3.5 we elucidate the effect of the trap aspect ratio — i.e., effective range — on the energetic and structural properties mentioned above. We conclude and discuss the relative merits of our approach in Section 4. Our work is strongly inspired by earlier, similar studies of trapped few-fermion systems subject to three-dimensional harmonic confinement — particularly Ref. [25], as well as Refs. [26–29]. These publications, in turn, are partly motivated by research on bosonic $^4$He and fermionic $^3$He droplets [30] which, although much denser than ultracold atomic gases, can be described using the same theoretical framework.

## 2  Model

The two-component Fermi gases considered in our analysis consist of equal-mass atoms with balanced spin populations, such that $N = N_\uparrow + N_\downarrow$ and $N_\uparrow = N_\downarrow = N/2$, where $N_\uparrow$ and $N_\downarrow$ denote the number of 'spin-up' and 'spin-down' fermions, respectively. Each gas is confined in an isotropic two-dimensional (2D) harmonic trap and in the non-interacting ground state only the first two harmonic oscillator shells are occupied — corresponding to particle numbers, $N_\uparrow + N_\downarrow = 1 + 1$, $2 + 2$, and $3 + 3$. Our work is inspired by recent experiments in the group of Selim Jochim [8,9], which show that the harmonically trapped ground state of a small number of fermionic $^6$Li atoms — ranging from 20 down to just 2 — can be prepared with very high fidelity.

The effective low-energy Hamiltonian reads as follows:

$$\mathcal{H} = \sum_{i=1}^{N} \left[ -\frac{\hbar^2}{2m} \nabla_{\mathbf{r}_i}^2 + V_{\text{ext}}(|\mathbf{r}_i|) \right] + \sum_{i<j}^{N} V_{\text{int}}(|\mathbf{r}_i - \mathbf{r}_j|) \,, \tag{1}$$

where $m$ is the atomic mass and $\mathbf{r}_i$ is the 2D position vector of the $i^{th}$ atom measured from the centre of the trap. The first term corresponds to the kinetic energy, the second term to the external confinement,

$$V_{\text{ext}}(|\mathbf{r}_i|) = \frac{m\omega_r^2}{2} r_i^2, \quad r_i \equiv |\mathbf{r}_i| \,, \tag{2}$$

where $\omega_r$ is the radial harmonic trapping frequency, and the third term to short-range pairwise interactions. Note that Pauli exclusion ensures identical fermions do not interact. The interactions between distinguishable fermions are described using a finite-range Gaussian potential, parameterised by a width $r_0$ ($> 0$) and a depth $V_0$ ($< 0$):

$$V_{\text{int}}(|\mathbf{r}|) = V_0 \exp\left(-\frac{r^2}{2r_0^2}\right) - V_0 \frac{r}{l_r} \exp\left[-\frac{r^2}{2(2r_0)^2}\right]. \tag{3}$$

Here, $l_r = \sqrt{\hbar/(m\omega_r)}$ is the radial harmonic oscillator length scale in the plane. This potential has previously been employed to model the breathing modes [24] and time-reversed pair

correlations [11] of a few interacting fermions in a 2D harmonic trap. In the non-interacting limit of $V_0 = 0$, the eigenvalues of the Hamiltonian (1) are $\varepsilon_{nm}^{(0)} = (2n + |m| + 1)\hbar\omega_r$, where $n = 0, 1, 2, \ldots$ is the principal quantum number and $m = 0, \pm 1, \pm 2, \ldots$ is the quantum number for orbital angular momentum.

The values of $r_0$ and $V_0$ can be adjusted to generate potentials with different $s$-wave scattering properties in 2D free space [31]. We solve the $s$-wave radial Schrödinger equation for the relative motion of two elastically scattering atoms, matching the logarithmic derivatives of the wave functions inside and outside the range of the interaction potential (3) to obtain the scattering phase shift $\delta(k)$. By subsequently fitting the phase shift to the known form [18–20] of its low-energy expansion in two dimensions,

$$\cot[\delta(k)] = \frac{2}{\pi}\left[\gamma + \ln\left(\frac{ka_{2D}}{2}\right)\right] + \frac{1}{\pi}k^2 r_{2D} + \ldots, \tag{4}$$

we ascertain both the $s$-wave scattering length $a_{2D}$ and effective range $r_{2D}$.[1] Above, $k \equiv |\mathbf{k}|$ is the magnitude of the relative wave vector between the atoms in the plane and $\gamma \simeq 0.577216$ is Euler's constant. With this definition of $a_{2D}$, the relative radial wave function has the logarithmic large-distance form $\psi(r) \propto \ln(a_{2D}/r)$ which is characteristic of zero-energy two-dimensional scattering. Importantly, the low-energy physics does not depend on the short-range details of the true interaction potential and is, instead, universally determined by $a_{2D}$ and $r_{2D}$. In all our calculations, we therefore choose Gaussian widths small enough ($r_0 \lesssim 0.1 l_r$) that higher order terms in the expansion (4) are negligible in the energy range of interest. In two dimensions a two-body bound state always exists — even for arbitrarily weak attractive interactions — since the scattering amplitude obtained by the analytic continuation of Eq. (4) to negative energies always exhibits a pole. In the zero-effective-range limit, the corresponding binding energy $\varepsilon_b$ is related to the 2D scattering length via $\varepsilon_b = 4\hbar^2 e^{-2\gamma}/(ma_{2D}^2)$. For finite $r_{2D}$ this relationship must be determined numerically from the phase shift expansion; however, $\varepsilon_b$ still serves as a monotonic proxy for interaction strength [32].

The scattering length is always positive ($a_{2D} > 0$) because it enters as the argument of the logarithm in Eq. (4) and the phase shift must remain real at low energies. In the many-body limit as $a_{2D}$ increases the two-component Fermi gas undergoes a crossover from a Bose–Einstein condensate (BEC) of tightly bound diatomic molecules to a Bardeen–Cooper–Schrieffer (BCS) superfluid of long-range Cooper pairs [32, 33]. However, unlike in three dimensions, there is no unitary limit where the interaction strength diverges and becomes scale invariant. Rather, the strongly interacting regime emerges around the point $\ln(k_F a_{2D}) = 0$, where the Fermi momentum $k_F$ determines the average interparticle spacing [32, 33]. In the few-body limit this spacing becomes ill-defined due to large fluctuations, making the regime of strong interactions more difficult to characterise for only a small number of atoms.

A two-dimensional geometry is experimentally realised by applying a strong harmonic confinement along the axial direction [8, 9], characterised by an angular frequency $\omega_z$ and a corresponding length scale $l_z = \sqrt{\hbar/(m\omega_z)}$. However, in reality, the gas extends a small but finite distance perpendicular to the plane. At low energy, when $l_z$ is small (such that $kl_z \ll 1$) but still much larger than the van der Waals range of the interactions, the two-body scattering of distinguishable fermions can be mapped to a purely 2D scattering amplitude with an effective range given by [21–24]

$$r_{2D} = -l_z^2 \ln(2). \tag{5}$$

As a result, the effect of a *quasi*-2D geometry on the scattering can be mimicked and probed by attributing a finite, negative value to the effective range in the 2D model, Eqs. (1)–(5).

---

[1]Note that the exact definitions of the 2D scattering length and effective range are not fixed in the literature. Our definition of $r_{2D}$ has units of squared length, consistent with Refs. [11, 24].

The effective range can be tuned through a wide range of negative values near a shape resonance [24, 34] which arises due to the structure of the model potential. Virtual bound states are supported in the attractive well associated with the first term of Eq. (3), and these can couple to free-space scattering states through the small repulsive barrier created by the second term. We restrict our calculations to the regime where this potential supports a single two-body $s$-wave bound state in two-dimensional free space [11, 24]. In Subsections 3.1–3.4 we fix the effective range to very nearly zero, $r_{2D}/l_r^2 = -0.001 \approx 0$, in order to determine the energetic and structural properties of the Fermi gases very close to the strictly 2D limit, which is of fundamental interest. Increasing $|r_{2D}|$ — while remaining within the regime of the mapping in Eq. (5) — leads to small quantitative shifts in these results but, most of the time, leaves them qualitatively unchanged. In Subsection 3.5 we show how our results are modified for $r_{2D}/l_r^2 = -0.2$ which was the largest negative value considered in Ref. [11].

We wish to emphasise that the two-Gaussian form of the interaction potential in Eq. (3) has been chosen for three reasons. First, it provides a minimal finite-range model that reproduces a target 2D scattering length $a_{2D}$, while the associated effective range $r_{2D}$ can be tuned over a wide range by adjusting the relative weights of the attractive well and repulsive barrier. Second, this potential is numerically tractable when complemented with the use of Gaussian basis functions. Third, it has been employed successfully in previous ECG calculations for quasi-2D few-fermion systems where it was shown to capture the relevant low-energy scattering properties [11, 24]. However, since the universal low-energy physics depends only on $a_{2D}$ and $r_{2D}$, and not on the microscopic details of the short-range potential, the *precise* functional form we use is unimportant. This was verified in Refs. [11, 24] where the modified potential given in Eq. (S23) of the Supplemental Material of Ref. [24] was shown to yield the same energies at fixed binding energy $\varepsilon_b$ and effective range.

## 3   Results and Discussion

To numerically solve the time-independent Schrödinger equation for the Hamiltonian (1) we employ the explicitly correlated Gaussian method discussed in detail in our earlier publication [11] (see Appendix A therein). Other works which have also applied this technique to study ultracold two-component fermions include Refs. [24–29]. Our calculations are parameterised in terms of the two-body binding energy $\varepsilon_b \geq 0$ and the effective range $r_{2D}$. Although $\varepsilon_b$ was introduced in Section 2 in the context of free-space pairwise scattering, it can additionally be defined in the presence of the harmonic trap. The two definitions coincide in the weak confinement limit and in both cases $\varepsilon_b$ remains a monotonic function of the underlying scattering parameters, $a_{2D}$ and $r_{2D}$. In practice, we determine the trapped value of $\varepsilon_b$ by using the ECG method to compute the relative ground-state energy $\varepsilon_{rel}$ for the $1+1$ system described by Eq. (1) with specified values of $r_0$ and $V_0$. The total ground-state energy in the harmonic trap is $\varepsilon = \varepsilon_{com} + \varepsilon_{rel} = 2\hbar\omega_r - \varepsilon_b$, and since the ground state contains no centre-of-mass excitations $\varepsilon_{com} = \hbar\omega_r$, we can immediately find $\varepsilon_b$.

### 3.1   Energy Spectra

In two dimensions the exact energy spectrum for $1+1$ fermions was analytically calculated by Busch et al. in 1998 [35]. Their approach involved modelling the interaction with a regularised Dirac delta distribution, expanding the relative wave function in the harmonic oscillator basis, and using standard integral representations to evaluate the Schrödinger equation. In 2010 Liu et al. numerically computed the exact energy spectrum for $2+1$ fermions by extending the approach of Efimov [36] to the two-dimensional trapped case and applying the Bethe–Peierls boundary condition [37]. In this subsection we obtain numerically exact energy

spectra for 1+1, 2+2, and 3+3 fermions at very nearly zero effective range, $r_{2D}/l_r^2 = -0.001$ $\approx 0$. After separating off the centre-of-mass degree of freedom, we expand the eigenstates of the relative Hamiltonian in terms of explicitly correlated Gaussian basis functions [11–15]. These basis functions depend on a series of non-linear variational parameters (the Gaussian widths) which are optimised by energy minimisation. In Fig. 1 we plot the resultant energies as functions of the two-body binding energy $\varepsilon_b$.

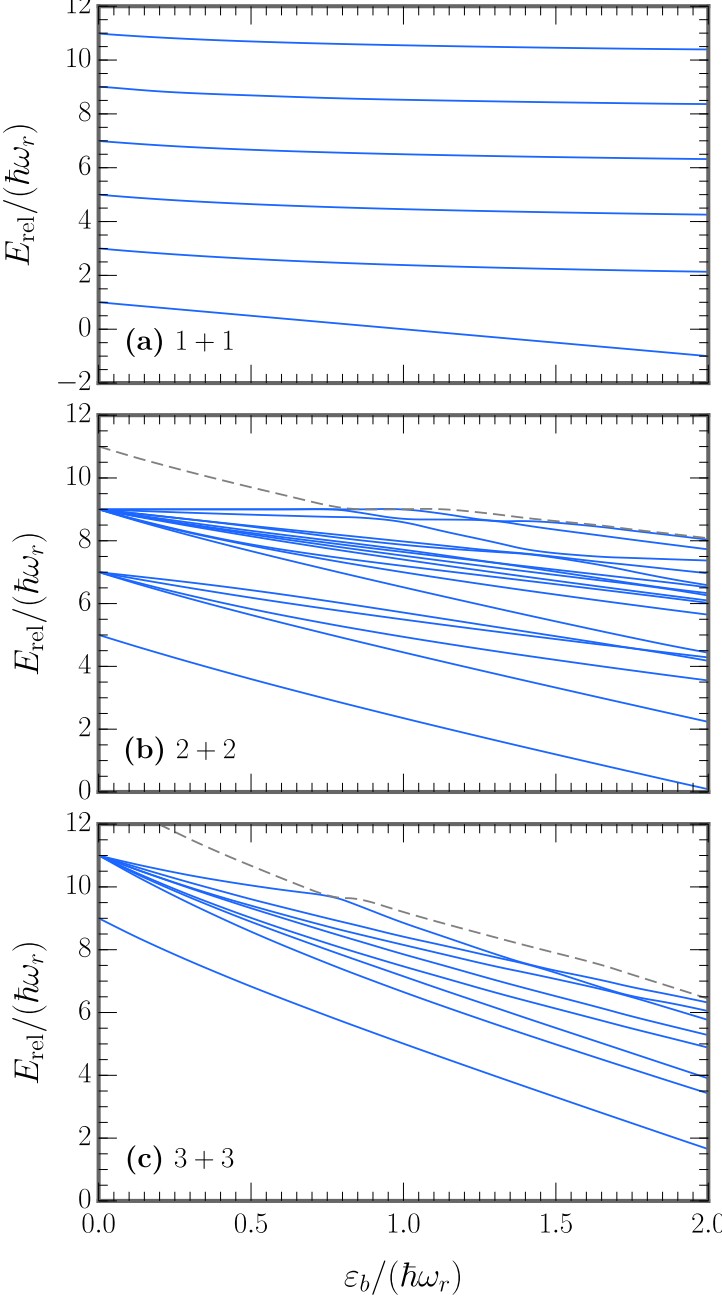

Figure 1: The monopole energy spectrum for (a) 1+1, (b) 2+2, and (c) 3+3 fermions at very nearly zero effective range, $r_{2D}/l_r^2 = -0.001 \approx 0$. $E_{rel}$ is the energy of relative motion and $\varepsilon_b$ is the two-body binding energy. In panels (b) and (c) the grey dashed line indicates the energy of the first state of the next (unshown) manifold.

The non-interacting ground state at $\varepsilon_b = 0$ can assume one of two configurations depend-
ing on the total number of particles $N$: either all of the degenerate single-particle states of
the highest energy level of the 2D harmonic oscillator are filled ('closed shell'), or some of
the degenerate states remain empty ('open shell'). The $1 + 1$ and $3 + 3$ systems both feature
a closed-shell ground state that is non-degenerate, whereas the $2 + 2$ ground state is open-
shell. We restrict our analysis to ground states characterised by zero total orbital angular mo-
mentum. For the $2 + 2$ system this means that the two highest energy opposite-spin fermions
reside in different degenerate single-particle states. Since the Hamiltonian is rotationally sym-
metric, only monopole excitations between states with the same (i.e., zero) total angular mo-
mentum occur. (The $m$ quantum numbers for all atoms sum to zero in both the ground and
excited states.) We can see in Fig. 1 that for all three atom numbers at $\varepsilon_b = 0$, all monopole
excitations have an energy of $2\hbar\omega_r$. This can be attributed either to exciting a single particle
up two harmonic oscillator shells, or to exciting a time-reversed pair of particles $(n, m, \uparrow)$ and
$(n, -m, \downarrow)$ up one shell each.

Our result for $1 + 1$ fermions in Fig. 1(a) agrees with the 'Busch spectrum' [35] for the con-
sidered range of binding energies, $0 \le \varepsilon_b \le 2\hbar\omega_r$. As evident in Fig. 2 of Ref. [11], this range
is sufficient to capture the non-monotonic dependence on $\varepsilon_b$ of the lowest monopole excitation
of $3 + 3$ fermions [Fig. 1(c)] — a feature which is driven by coherent pair correlations [38].
Larger basis sizes are required for the ECG method to converge at higher binding energies,
$\varepsilon_b > 2\hbar\omega_r$, where the tight composite bosonic wave functions become difficult to represent
numerically [11, 25]. Currently, convergence cannot be achieved in this regime for six atoms,
although it may be possible for four (and is certainly possible for two). It is additionally chal-
lenging to solve for more than six particles at *any* binding energy due to the factorial growth
(with $N$) in the number of permutations of identical fermions required to antisymmetrise the
full wave function [11, 25]. The spectra in Fig. 1 for increasing $N$ are qualitatively similar, but
increasingly complex due to the existence of higher degeneracies in the non-interacting limit.
For $1 + 1$ fermions [Fig. 1(a)] we choose to show the six lowest energy states, while for $2 + 2$
fermions [Fig. 1(b)] we choose to show the ground state and the first- and second-excited-state
manifolds. For $3 + 3$ fermions [Fig. 1(c)] we display the ground state and the first-excited-state
manifold which, in this case, is the largest number of states that can be computed to numerical
convergence within a reasonable time frame (on the order of months).

## 3.2 Density Matrices and Occupation Numbers

### 3.2.1 One-Body Density Matrix

In the first-quantised position representation the one-body density matrix for the spin-$\uparrow$ parti-
cles is given by

$$\rho_\uparrow(\mathbf{r}, \mathbf{r}') = \left[ \int \cdots \int d\mathbf{r}_1^\uparrow d\mathbf{r}_2^\downarrow \cdots d\mathbf{r}_{N-1}^\uparrow d\mathbf{r}_N^\downarrow \left| \Psi(\mathbf{r}_1^\uparrow, \mathbf{r}_2^\downarrow, \cdots, \mathbf{r}_{N-1}^\uparrow, \mathbf{r}_N^\downarrow) \right|^2 \right]^{-1} \times$$

$$\int \cdots \int d\mathbf{r}_2^\downarrow d\mathbf{r}_3^\uparrow d\mathbf{r}_4^\downarrow \cdots d\mathbf{r}_{N-1}^\uparrow d\mathbf{r}_N^\downarrow \Psi(\mathbf{r}, \mathbf{r}_2^\downarrow, \mathbf{r}_3^\uparrow, \mathbf{r}_4^\downarrow, \cdots, \mathbf{r}_{N-1}^\uparrow, \mathbf{r}_N^\downarrow) \Psi^*(\mathbf{r}', \mathbf{r}_2^\downarrow, \mathbf{r}_3^\uparrow, \mathbf{r}_4^\downarrow, \cdots, \mathbf{r}_{N-1}^\uparrow, \mathbf{r}_N^\downarrow),$$

$$\tag{6}$$

where $\Psi$ is the total $N$-body wave function and all integrals are two-dimensional ($d\mathbf{r} \equiv d^2\mathbf{r}$).
The first line above is a normalisation constant; in the second line the density $\Psi\Psi^*$ is integrated
over all co-ordinates except those of a single spin-$\uparrow$ atom.

The matrix elements of Eq. (6) in the explicitly correlated Gaussian basis were derived in
our earlier work (see Appendices A, C, and D of Ref. [11]); for ease of reference, we quote the

final result below:

$$[\rho_\uparrow(\mathbf{r}, \mathbf{r}')]_{\mathbb{A}\mathbb{A}'} \equiv \langle \phi_{\mathbb{A}} | \rho_\uparrow(\mathbf{r}, \mathbf{r}') | \phi_{\mathbb{A}'} \rangle = c_1 \exp\left\{ -\frac{1}{2}\left[ c\mathbf{r}^2 + c'(\mathbf{r}')^2 - a\mathbf{r}^T\mathbf{r}' \right] \right\}, \tag{7}$$

which contains the following scalars:

$$c_1 = \frac{(2\pi)^{N-1}}{\det[\mathbb{B} + \mathbb{B}']}, \tag{8a}$$

$$c = b_1 - \mathbf{b}^T\mathbb{C}\mathbf{b}, \tag{8b}$$

$$c' = b_1' - (\mathbf{b}')^T\mathbb{C}\mathbf{b}', \tag{8c}$$

$$a = \mathbf{b}^T\mathbb{C}\mathbf{b}' + (\mathbf{b}')^T\mathbb{C}\mathbf{b}. \tag{8d}$$

Here, $b_1 = (\mathbb{U}^T\mathbb{A}\mathbb{U})_{11}$ is also a scalar, $\mathbf{b} = ((\mathbb{U}^T\mathbb{A}\mathbb{U})_{12}, \ldots, (\mathbb{U}^T\mathbb{A}\mathbb{U})_{1N})$ is an $(N-1)$-dimensional vector, $\mathbb{B}$ is an $(N-1) \times (N-1)$-dimensional matrix given by $\mathbb{U}^T\mathbb{A}\mathbb{U}$ with the first row and column removed, and $\mathbb{C} = (\mathbb{B} + \mathbb{B}')^{-1}$. The $N \times N$ transformation matrix $\mathbb{U}$ ($\mathbf{x} = \mathbb{U}\mathbf{y}$) converts the single-particle co-ordinates $\mathbf{y}$ into relative and centre-of-mass generalised Jacobi co-ordinates $\mathbf{x}$ (where $\mathbf{x}$ and $\mathbf{y}$ are vectors of vectors). The $N \times N$ correlation matrix $\mathbb{A}$ comprises non-linear variational parameters (the Gaussian widths) which are optimised stochastically. Operationally, the stochastic variational procedure proposes random updates to the elements of $\mathbb{A}$ and retains them only if they lower the variational ground-state energy. The bounds of these random proposals are chosen to reflect the physically relevant interparticle length scales, which differ for distinguishable and indistinguishable fermions due to the Pauli principle. Each ECG basis function $|\phi_{\mathbb{A}}\rangle$ is numerically represented by a unique $\mathbb{A}$ matrix [11].

Equation (6) can be expanded over a complete set of basis functions — the natural orbitals $\chi_{nm}(\mathbf{r})$ — where the expansion coefficients correspond to the occupation numbers $\mathcal{N}_{nm}$ of those orbitals:

$$\rho_\uparrow(\mathbf{r}, \mathbf{r}') = \sum_{nm} \mathcal{N}_{nm}\, \chi_{nm}^*(\mathbf{r})\, \chi_{nm}(\mathbf{r}'). \tag{9}$$

These components are normalised as follows:

$$\int d\mathbf{r}\, \chi_{nm}^*(\mathbf{r})\, \chi_{n'm'}(\mathbf{r}) = \delta_{nn'}\delta_{mm'}, \tag{10a}$$

$$\sum_{nm} \mathcal{N}_{nm} = 1, \tag{10b}$$

where $(n, m)$ are the harmonic oscillator quantum numbers defined below Eq. (3), and where the asterisk denotes complex conjugation (although in our specific case the natural orbitals are real). This natural-orbital decomposition of the one-body density matrix follows the standard framework introduced by Löwdin [39] in the context of quantum chemistry, and independently by Penrose and Onsager [40] in the context of Bose–Einstein condensation. Yang [41] further developed the framework by formulating the criterion of off-diagonal long-range order. This approach has since been widely adopted in ultracold-atom physics, including analyses of interacting many-body Bose systems by DuBois and Glyde [42] and few-body studies of trapped Bose gases by Zöllner et al. [43]. In these applications the eigenvalues (i.e., occupation numbers) of the one-body density matrix provide a basis-independent characterisation of single-particle structure and condensation.

In practice, because direct decomposition of the four-dimensional object $\rho_\uparrow(\mathbf{r}, \mathbf{r}')$ in the form of Eq. (9) is computationally infeasible, we first reduce the number of degrees of freedom by defining partial-wave projections:

$$\rho_\uparrow^m(r, r') = \frac{1}{2\pi} \int_0^{2\pi} \int_0^{2\pi} d\theta\, d\theta'\, e^{-im\theta} \rho_\uparrow(\mathbf{r}, \mathbf{r}')\, e^{im\theta'}, \tag{11}$$

with $\theta^{(\prime)}$ denoting the angle associated with the vector $\mathbf{r}^{(\prime)}$ and $r^{(\prime)} \equiv |\mathbf{r}^{(\prime)}|$. This procedure mirrors that used in the few-body fermionic studies of Blume and Daily [25] where the three-dimensional case is addressed. The explicitly correlated Gaussian matrix elements of Eq. (11) are

$$[\rho_\uparrow^m(r, r')]_{\mathbb{A}\mathbb{A}'} \equiv \langle \phi_\mathbb{A} | \rho_\uparrow^m | \phi_{\mathbb{A}'} \rangle = 2\pi \, c_1 \, \mathcal{I}_m\left(\frac{a r r'}{2}\right) \exp\left\{-\frac{1}{2}\Big[c r^2 + c'(r')^2\Big]\right\}, \qquad (12)$$

where $\mathcal{I}_m(x)$ is the modified Bessel function of the first kind, and where the scalars $\{c_1, c, c', a\}$ have been defined in Eq. (8).

The ground-state ('GS') matrix element of the projected one-body density matrix can now be written as

$$[\rho_\uparrow^m(r, r')]_{\text{GS}} \equiv \frac{\langle \Psi^{(\text{GS})} | \rho_\uparrow^m(r, r') | \Psi^{(\text{GS})} \rangle}{\langle \Psi^{(\text{GS})} | \Psi^{(\text{GS})} \rangle} = \frac{\sum_{i,j} c_i^* [\rho_\uparrow^m(r, r')]_{\mathbb{A}_i \mathbb{A}_j} c_j}{\sum_{i,j} c_i^* \mathbb{O}_{\mathbb{A}_i \mathbb{A}_j} c_j} \, . \qquad (13)$$

Above, the second expression is obtained from the first by inserting two complete sets of explicitly correlated Gaussian basis states into both the numerator and denominator. The $i^{th}$ (real) coefficient of the total ground-state wave function in this basis is $c_i \equiv \langle \phi_{\mathbb{A}_i} | \Psi^{(\text{GS})} \rangle$, and the overlap matrix element is [14]

$$\mathbb{O}_{\mathbb{A}_i \mathbb{A}_j} \equiv \langle \phi_{\mathbb{A}_i} | \phi_{\mathbb{A}_j} \rangle = \frac{(2\pi)^N}{\det[\mathbb{A}_i + \mathbb{A}_j]} \, . \qquad (14)$$

The indices $i$ and $j$ both run over the minimum number of (previously found) optimised basis states required to converge the ground-state energy at a given two-body binding energy $\varepsilon_b$. While the equations in this and later subsections are written in terms of unsymmetrised basis states for clarity, these must be antisymmetrised to account for particle exchange (refer to Appendix D of Ref. [11] for further details).

At this point, the occupation numbers can be found by discretising the variables $r$ and $r'$ into grids of width $\Delta r$ and then finding the eigenvalues of $\sqrt{r}\,[\rho_\uparrow^m(r, r')]_{\text{GS}}\sqrt{r'}\Delta r$ for a given partial wave $m$. The first such eigenvalue is $\mathcal{N}_{n=0,m}$, the second is $\mathcal{N}_{n=1,m}$, and so on. These results are shown in panels (a), (c), and (e) of Fig. 2. In the non-interacting limit of $\varepsilon_b = 0$, where the natural orbitals are the single-particle harmonic oscillator levels, they are straightforward to understand. Due to the antisymmetry of the wave function same-spin fermions must occupy different single-particle levels. For $1+1$ fermions the spin-up atom is in the $n = m = 0$ ground state, which has an occupation number of $\mathcal{N}_{0,0} = 1$ due to the normalisation condition (10b), while all other occupation numbers are zero. For $2+2$ fermions the second spin-up atom is equally distributed between the two degenerate first excited states with $n = 0$ and $m = \pm 1$ — leading to three finite occupation numbers, $\mathcal{N}_{0,0} = 1/2$ and $\mathcal{N}_{0,\pm 1} = 1/4$. In the $3+3$ case, the three lowest energy states contain one spin-up fermion each and thus the corresponding occupation numbers become $\mathcal{N}_{0,0} = \mathcal{N}_{0,\pm 1} = 1/3$, whereas all others vanish.

When the binding energy increases ($\varepsilon_b > 0$) the finite values of $\mathcal{N}_{0,0}$ and $\mathcal{N}_{0,\pm 1}$ decrease, while the occupation numbers of higher excited natural orbitals increase as one would generally expect. However, for the range of interaction strengths covered by the energy spectra in Subsection 3.1 ($0 \leq \varepsilon_b \leq 2\hbar\omega_r$) this variation is not strong — and the one-body density matrix can always be decomposed with a good level of accuracy by only including up to six natural orbitals. Such an observation suggests that we are never close to the deep Bose–Einstein condensation regime. If we instead had a tight composite bosonic wave function, then its expansion into effective single-particle orbitals (the natural orbitals of $\rho_\uparrow$) would require many terms [25]. In that case, many more occupation numbers would take on (small but) non-vanishing values, forcing a more significant reduction in the values of $\mathcal{N}_{0,0}$ and $\mathcal{N}_{0,\pm 1}$ than what can be seen in Fig. 2.

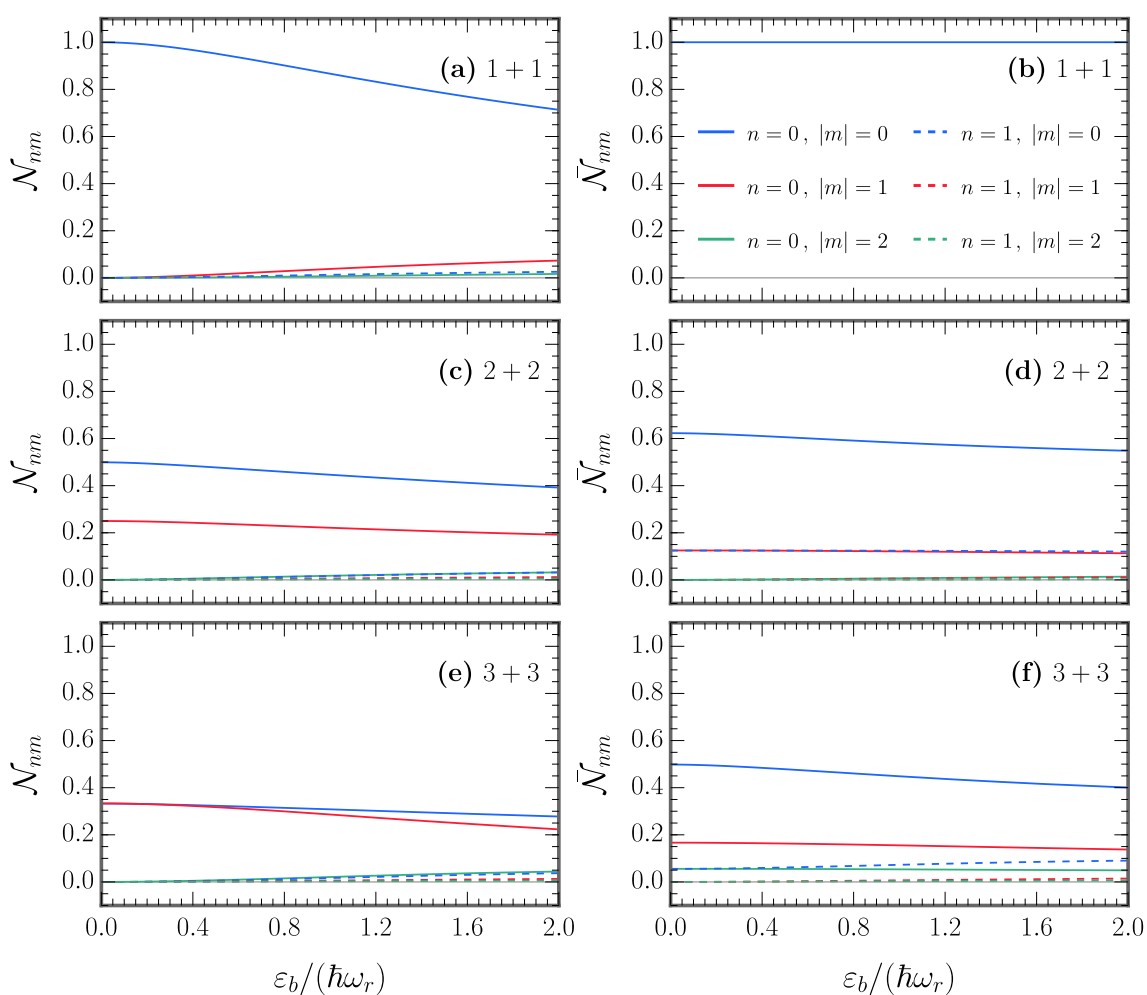

Figure 2: Left panels: Ground-state occupation numbers (eigenvalues) of the one-body density matrix $\mathcal{N}_{nm}$ (9) for (a) $1+1$, (c) $2+2$, and (e) $3+3$ fermions. Right panels: Ground-state occupation numbers of the reduced two-body density matrix $\bar{\mathcal{N}}_{nm}$ (17) for (b) $1+1$, (d) $2+2$, and (f) $3+3$ fermions. The results are plotted as a function of the two-body binding energy $\varepsilon_b$ for (very nearly) zero effective range, $r_{2D}/l_r^2 = -0.001 \approx 0$. Note in panel (b) that for any binding energy $\bar{\mathcal{N}}_{0,0} = 1$, while all other occupation numbers vanish.

### 3.2.2  Two-Body Density Matrix

The two-body density matrix in the first-quantised position representation is given by

$$
\rho(\mathbf{r}_1, \mathbf{r}_1'; \mathbf{r}_2, \mathbf{r}_2') = \left[ \int \cdots \int d\mathbf{r}_1^\uparrow d\mathbf{r}_2^\downarrow \cdots d\mathbf{r}_{N-1}^\uparrow d\mathbf{r}_N^\downarrow \left| \Psi(\mathbf{r}_1^\uparrow, \mathbf{r}_2^\downarrow, \cdots, \mathbf{r}_{N-1}^\uparrow, \mathbf{r}_N^\downarrow) \right|^2 \right]^{-1} \times
$$

$$
\int \cdots \int d\mathbf{r}_3^\uparrow d\mathbf{r}_4^\downarrow \cdots d\mathbf{r}_{N-1}^\uparrow d\mathbf{r}_N^\downarrow \Psi(\mathbf{r}_1, \mathbf{r}_2, \mathbf{r}_3^\uparrow, \mathbf{r}_4^\downarrow, \cdots, \mathbf{r}_{N-1}^\uparrow, \mathbf{r}_N^\downarrow) \Psi^*(\mathbf{r}_1', \mathbf{r}_2', \mathbf{r}_3^\uparrow, \mathbf{r}_4^\downarrow, \cdots, \mathbf{r}_{N-1}^\uparrow, \mathbf{r}_N^\downarrow),
$$

$$(15)$$

where the density $\Psi\Psi^*$ is integrated over all co-ordinates except those of one spin-$\uparrow$ particle and one spin-$\downarrow$ particle. In two dimensions $\rho(\mathbf{r}_1, \mathbf{r}_1'; \mathbf{r}_2, \mathbf{r}_2')$ is an eight-dimensional array, so

we again need to reduce the number of degrees of freedom prior to diagonalisation. To this end we follow Ref. [25], which considered the three-dimensional version of this problem, and transform from the co-ordinates of the individual atoms to the centre-of-mass and relative co-ordinates of the two spin-↑-spin-↓ pairs: $\mathbf{R} = (\mathbf{r}_1 + \mathbf{r}_2)/2$ and $\mathbf{r} = \mathbf{r}_1 - \mathbf{r}_2$ (and their primed equivalents). By setting $\mathbf{r} = \mathbf{r}'$ we can then define the *reduced* two-body density matrix as

$$\rho_{\text{red}}(\mathbf{R}, \mathbf{R}') = \int d\mathbf{r}\, \rho\left(\mathbf{R} + \frac{\mathbf{r}}{2}, \mathbf{R}' + \frac{\mathbf{r}}{2}; \mathbf{R} - \frac{\mathbf{r}}{2}, \mathbf{R}' - \frac{\mathbf{r}}{2}\right), \tag{16}$$

which measures non-local correlations between pairs described by the *same* relative-distance vector.

In analogy to the one-body density matrix, the reduced two-body density matrix can be expanded in terms of natural orbitals and occupation numbers:

$$\rho_{\text{red}}(\mathbf{R}, \mathbf{R}') = \sum_{nm} \bar{\mathcal{N}}_{nm}\, \bar{\chi}_{nm}^*(\mathbf{R})\, \bar{\chi}_{nm}(\mathbf{R}'), \tag{17}$$

which have the normalisations,

$$\int d\mathbf{R}\, \bar{\chi}_{nm}^*(\mathbf{R})\, \bar{\chi}_{n'm'}(\mathbf{R}) = \delta_{nn'}\, \delta_{mm'}, \tag{18a}$$

$$\sum_{nm} \bar{\mathcal{N}}_{nm} = 1. \tag{18b}$$

We again perform partial-wave projections according to Eq. (11): $\rho_{\text{red}}(\mathbf{R}, \mathbf{R}') \to \rho_{\text{red}}^m(R, R')$ with $R^{(\prime)} \equiv |\mathbf{R}^{(\prime)}|$. The derivation of the ground-state matrix element of the *projected reduced two-body density matrix* $[\rho_{\text{red}}^m(R, R')]_{\text{GS}}$ then follows identically to Eqs. (12)–(13) with only one minor change. The vector of single-particle co-ordinates $\mathbf{y}$ must be replaced by $\mathbf{y}'$,

$$\mathbf{y} = (\mathbf{r}_1^\uparrow, \mathbf{r}_2^\downarrow, \mathbf{r}_3^\uparrow, \ldots, \mathbf{r}_N^\downarrow) \to \mathbf{y}' = (\mathbf{R}, \mathbf{r}, \mathbf{r}_3^\uparrow, \ldots, \mathbf{r}_N^\downarrow), \tag{19}$$

and therefore the transformation matrix $\mathbb{U}$ should be redefined appropriately, $\mathbf{x} = \mathbb{U}'\mathbf{y}'$ [25]. The replacement matrix $\mathbb{U}'$ that takes the place of $\mathbb{U}$ is shown below for each of the total particle numbers $(N_\uparrow + N_\downarrow)$ considered in this work; for reference, the original $\mathbb{U}$ matrices were defined in Eq. (A.2) of Ref. [11]:

$$1+1: \quad \mathbb{U} = \begin{pmatrix} 1 & -1 \\ \frac{1}{2} & \frac{1}{2} \end{pmatrix} \to \mathbb{U}' = \begin{pmatrix} 0 & 1 \\ 1 & 0 \end{pmatrix}, \tag{20a}$$

$$2+2: \quad \mathbb{U} = \begin{pmatrix} 1 & -1 & 0 & 0 \\ 0 & 0 & 1 & -1 \\ \frac{1}{2} & \frac{1}{2} & -\frac{1}{2} & -\frac{1}{2} \\ \frac{1}{4} & \frac{1}{4} & \frac{1}{4} & \frac{1}{4} \end{pmatrix} \to \mathbb{U}' = \begin{pmatrix} 0 & 1 & 0 & 0 \\ 0 & 0 & 1 & -1 \\ 1 & 0 & -\frac{1}{2} & -\frac{1}{2} \\ \frac{1}{2} & 0 & \frac{1}{4} & \frac{1}{4} \end{pmatrix}, \tag{20b}$$

$$3+3: \quad \mathbb{U} = \begin{pmatrix} 1 & -1 & 0 & 0 & 0 & 0 \\ 0 & 0 & 1 & -1 & 0 & 0 \\ 0 & 0 & 0 & 0 & 1 & -1 \\ \frac{1}{2} & \frac{1}{2} & -\frac{1}{2} & -\frac{1}{2} & 0 & 0 \\ \frac{1}{4} & \frac{1}{4} & \frac{1}{4} & \frac{1}{4} & -\frac{1}{2} & -\frac{1}{2} \\ \frac{1}{6} & \frac{1}{6} & \frac{1}{6} & \frac{1}{6} & \frac{1}{6} & \frac{1}{6} \end{pmatrix} \to \mathbb{U}' = \begin{pmatrix} 0 & 1 & 0 & 0 & 0 & 0 \\ 0 & 0 & 1 & -1 & 0 & 0 \\ 0 & 0 & 0 & 0 & 1 & -1 \\ 1 & 0 & -\frac{1}{2} & -\frac{1}{2} & 0 & 0 \\ \frac{1}{2} & 0 & \frac{1}{4} & \frac{1}{4} & -\frac{1}{2} & -\frac{1}{2} \\ \frac{1}{3} & 0 & \frac{1}{6} & \frac{1}{6} & \frac{1}{6} & \frac{1}{6} \end{pmatrix}. \tag{20c}$$

The occupation numbers $\bar{\mathcal{N}}_{nm}$ are obtained as the eigenvalues of $\sqrt{R}\,[\rho_{\text{red}}^m(R, R')]_{\text{GS}}\sqrt{R'}$ $\times \Delta R$ and are displayed in panels (b), (d), and (f) of Fig. 2. Although the values in the non-interacting limit ($\varepsilon_b = 0$) are less intuitive than in the one-body case, they may be verified

by comparing against analytically derived results. In Appendix A we detail these steps for the $2 + 2$ system as an example. For increasing binding energy ($\varepsilon_b > 0$) the occupation numbers from the reduced two-body density matrix follow the same qualitative trends as those from the one-body density matrix. It may initially seem counter-intuitive that the largest eigenvalue $\bar{\mathcal{N}}_{0,0}$ has a higher value in the absence of pairs ($\varepsilon_b = 0$) than in the presence of pairs ($\varepsilon_b \gg 0$). However, this is directly due to the procedure used to eliminate degrees of freedom and define the quantity $\rho_{\text{red}}(\mathbf{R}, \mathbf{R}')$ — and was similarly observed in the three-dimensional case [25].

### 3.2.3 Molecular Condensate Fraction

In a trapped one-component Bose gas the condensate fraction becomes appreciable when the lowest eigenvalue of the one-body density matrix becomes of order unity. In a two-component Fermi gas, by contrast, none of the natural orbitals of the one-body density matrix can become macroscopically occupied due to the antisymmetry of the wave function under particle exchange. A significant condensate fraction can only arise when bosonic pairs are formed, and hence, such insight must instead come from an analysis of the two-body density matrix.

Due to the elimination of degrees of freedom as described above, the absolute magnitude of $\bar{\mathcal{N}}_{0,0}$ no longer corresponds directly to the number of condensed pairs. Rather, condensation occurs when the lowest natural orbital of the reduced two-body density matrix becomes macroscopically occupied — in other words, when $\bar{\mathcal{N}}_{0,0}$ greatly exceeds all other $\bar{\mathcal{N}}_{nm}$ [25]. Accordingly, we define the condensate fraction $\mathcal{N}_{\text{cond}}$ in two dimensions as follows:

$$\mathcal{N}_{\text{cond}} = 1 - \frac{\max\left(\sum_{m=\pm\dots} \bar{\mathcal{N}}_{nm}\right)}{\bar{\mathcal{N}}_{0,0}} \left[(n, m) \neq (0, 0)\right], \qquad (21)$$

i.e., one minus 'the largest competing eigenvalue divided by the $(0, 0)$ eigenvalue' of the reduced two-body density matrix. Notice that the sum applies to non-zero $m$: when $m = 0$ the physical mode is unique and its occupation is the single value $\bar{\mathcal{N}}_{n,0}$; when $|m| > 0$ the physical mode is the multiplet with angular momentum $|m|$, consisting of the two degenerate states $+m$ and $-m$, so the relevant quantity to compare is the total occupation of that multiplet. In the deep molecular regime essentially all pairs occupy a single two-body natural orbital, so $\bar{\mathcal{N}}_{0,0}$ becomes much larger than any other $\bar{\mathcal{N}}_{nm}$ and the ratio in Eq. (21) correspondingly becomes very small: $\mathcal{N}_{\text{cond}} \to 1$. In the non-interacting limit ($\varepsilon_b = 0$) several $\bar{\mathcal{N}}_{nm}$ have comparable magnitude, so the 'largest competitor' in Eq. (21) is of the same order as $\bar{\mathcal{N}}_{0,0}$. The ratio in Eq. (21) is thus of order unity and $\mathcal{N}_{\text{cond}}$ becomes small: it approaches zero in the many-body limit, while for few-body systems it settles to a finite fraction less than one. We remark that Eq. (21) is directly analogous to the three-dimensional definition given in Eq. (16) of Ref. [25].

The eigenvalues $\bar{\mathcal{N}}_{nm}$ measure the extent to which the $N$-body state comprises pairs with centres of mass occupying the two-body natural orbital labelled by $(n, m)$. As the binding energy $\varepsilon_b$ is increased, the reduced two-body density matrix redistributes its fixed total spectral weight: the dominant eigenvalues decrease in absolute magnitude, while many additional eigenvalues 'turn on' from zero yet remain extremely small. These small components reflect the weak occupation of non-condensed and excited centre-of-mass pair configurations that appear as the interactions become stronger. The behaviour of the condensate fraction is determined by the relative evolution of the leading ($\bar{\mathcal{N}}_{0,0}$) and subleading eigenvalues: $\mathcal{N}_{\text{cond}}$ increases when $\bar{\mathcal{N}}_{0,0}$ falls off more slowly than the subleading eigenvalue, and decreases when the opposite holds. Therefore, $\mathcal{N}_{\text{cond}}$ grows when the *relative* dominance of $\bar{\mathcal{N}}_{0,0}$ over the rest of the spectrum increases. Equation (21) permits the identity of the subleading mode to change as the interaction strength varies; however, as seen in Fig. 2 the same mode remains subleading throughout the entire range of binding energies considered.

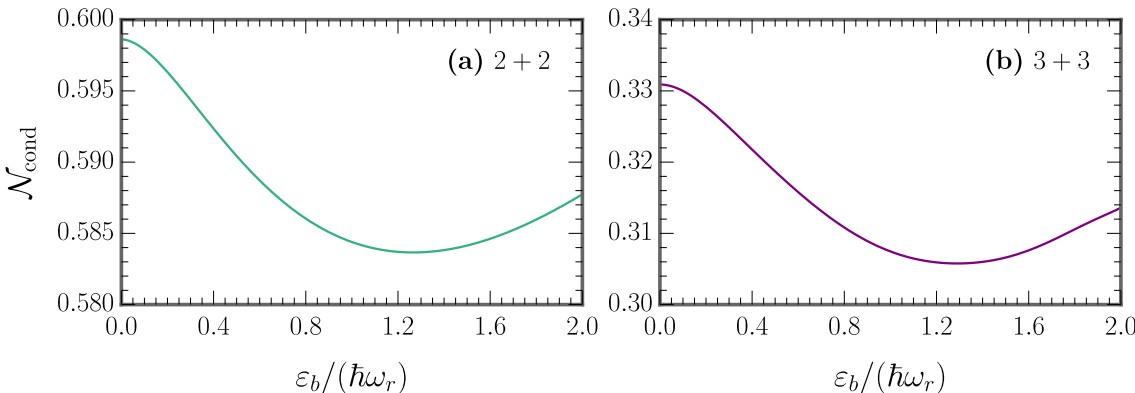

Figure 3: The condensate fraction $\mathcal{N}_{\mathrm{cond}}$ (22) as a function of the two-body binding energy $\varepsilon_b$ for (a) $2+2$ and (b) $3+3$ fermions in the ground state. The effective range is very close to zero in each panel, $r_{2D}/l_r^2 = -0.001 \approx 0$.

369    In Fig. 3 we plot the condensate fractions for the $2+2$ and $3+3$ Fermi systems as functions
370    of the interaction strength:

$$\mathcal{N}_{\mathrm{cond}} = 1 - \frac{\bar{\mathcal{N}}_{0,+1} + \bar{\mathcal{N}}_{0,-1}}{\bar{\mathcal{N}}_{0,0}} . \tag{22}$$

371    In both cases the behaviour is qualitatively similar. For $\varepsilon_b \gtrsim 1.2\hbar\omega_r$ the condensate fraction
372    shows a gentle upward trend, consistent with a gradual strengthening of pairing correlations.
373    For $\varepsilon_b \lesssim 1.2\hbar\omega_r$, however, the condensate fraction initially decreases before rising again, re-
374    sulting in a weakly non-monotonic dependence over the range $\varepsilon_b \lesssim 2\hbar\omega_r$. This behaviour ar-
375    ises because at weaker binding the dominant eigenvalue decreases more rapidly than its near-
376    est competitor, while at stronger binding the relative rates reverse. It is important to note that
377    the total variation on the vertical axis is very small, which means that the condensate fraction
378    is effectively flat on the non-molecular side of the two-dimensional crossover. This observation
379    is consistent with results from a complementary three-dimensional study. In particular, Fig. 11
380    of Ref. [25] shows that in the corresponding regime of negative inverse scattering length in
381    three dimensions, the condensate fractions of the $2+1$, $2+2$, and $3+2$ Fermi systems likewise
382    remain very nearly constant (for the $3+3$ system only two data points are available, so no ov-
383    erall trend can be inferred for that case).
384    We do not extend our analysis to larger binding energies $\varepsilon_b$ where the condensate fraction
385    approaches unity because, in practice, the explicitly correlated Gaussian method becomes in-
386    creasingly difficult to converge in this regime. As the binding energy grows, the internal size of
387    each $\uparrow\downarrow$ pair shrinks and the relative wave function develops structure on progressively shorter
388    length scales. Accurately resolving these sharper features requires Gaussians with very small
389    widths, while the overall trapped state still demands basis functions with much larger spatial
390    extent. For systems with more than two atoms this separation of length scales rapidly ampli-
391    fies the number of basis functions required for convergence. This challenge is compounded by
392    the stochastic nature of the basis-optimisation process, which involves generating and testing
393    many candidate basis functions at each expansion step, further increasing the computational
394    burden. In the $3+3$ case the computational cost becomes prohibitive before a tightly bound
395    molecular regime is reached, and even for $2+2$ the basis sizes needed at higher $\varepsilon_b$ are substan-
396    tially larger than those required in the crossover regime. For this reason, our results focus on

an intermediate range of binding energies for which fully converged calculations are attainable across all particle numbers considered.

### 3.3  Momentum Distributions

The momentum distribution of the spin-↑ atoms is given by the Fourier transform of the one-body density matrix defined in Eq. (6):

$$
n_\uparrow(\mathbf{k}) = \frac{1}{(2\pi)^2} \int \int d\mathbf{r}\, d\mathbf{r}'\, \rho_\uparrow(\mathbf{r}, \mathbf{r}') \exp\left[-i\mathbf{k}^T(\mathbf{r} - \mathbf{r}')\right]. \tag{23}
$$

It is straightforward to prove that Eq. (23) is equivalent to

$$
n_\uparrow(\mathbf{k}) = \sum_{nm} \mathcal{N}_{nm} |\widetilde{\chi}_{nm}(\mathbf{k})|^2, \tag{24}
$$

where

$$
\widetilde{\chi}_{nm}(\mathbf{k}) = \frac{1}{2\pi} \int d\mathbf{r}\, \chi_{nm}(\mathbf{r}) \exp\left(-i\mathbf{k}^T\mathbf{r}\right) \tag{25}
$$

is the Fourier transform of the natural orbitals introduced in Eq. (9). In order to obtain an analytical expression for the matrix elements of Eq. (23) in the explicitly correlated Gaussian basis, we can use the result for $[\rho_\uparrow(\mathbf{r}, \mathbf{r}')]_{\mathbb{A}\mathbb{A}'}$ shown in Eq. (7):

$$
[n_\uparrow(\mathbf{k})]_{\mathbb{A}\mathbb{A}'} = \frac{c_1}{(2\pi)^2} \int \int d\mathbf{r}\, d\mathbf{r}' \exp\left\{-\frac{1}{2}\left[c\mathbf{r}^2 + c'(\mathbf{r}')^2 - a\mathbf{r}^T\mathbf{r}'\right]\right\} \exp\left[i\mathbf{k}^T(\mathbf{r}' - \mathbf{r})\right]. \tag{26}
$$

By defining $\mathbf{X} = \mathbf{r}' - \mathbf{r}$ the equation above becomes

$$
[n_\uparrow(\mathbf{k})]_{\mathbb{A}\mathbb{A}'} = \frac{c_1}{(2\pi)^2} \int \int d\mathbf{r}\, d\mathbf{X} \exp\left[\frac{1}{2}\left(g_1\mathbf{r}^2 + g_2\mathbf{X}^2 + g_3\mathbf{r}^T\mathbf{X}\right)\right] \exp\left(i\mathbf{k}^T\mathbf{X}\right), \tag{27}
$$

involving the coefficients,

$$
g_1 = a - c - c', \tag{28a}
$$
$$
g_2 = -c', \tag{28b}
$$
$$
g_3 = a - 2c'. \tag{28c}
$$

The integral over $\mathbf{r}$ can be performed analytically for $g_1 < 0$:

$$
[n_\uparrow(\mathbf{k})]_{\mathbb{A}\mathbb{A}'} = -\frac{c_1}{2\pi g_1} \int d\mathbf{X} \exp\left(\frac{1}{2}g_4\mathbf{X}^2\right) \exp\left(i\mathbf{k}^T\mathbf{X}\right), \tag{29}
$$

with the coefficient defined as

$$
g_4 = g_2 - g_3^2/(4g_1). \tag{30}
$$

Subsequently, the integral over $\mathbf{X}$ can be analytically carried out for $g_4 < 0$:

$$
[n_\uparrow(\mathbf{k})]_{\mathbb{A}\mathbb{A}'} \equiv [n_\uparrow(k)]_{\mathbb{A}\mathbb{A}'} = \frac{c_1}{g_1 g_4} \exp\left(\frac{1}{2g_4}k^2\right), \quad k \equiv |\mathbf{k}|, \tag{31}
$$

where the coefficient $c_1/(g_1 g_4)$ can be either positive or negative.

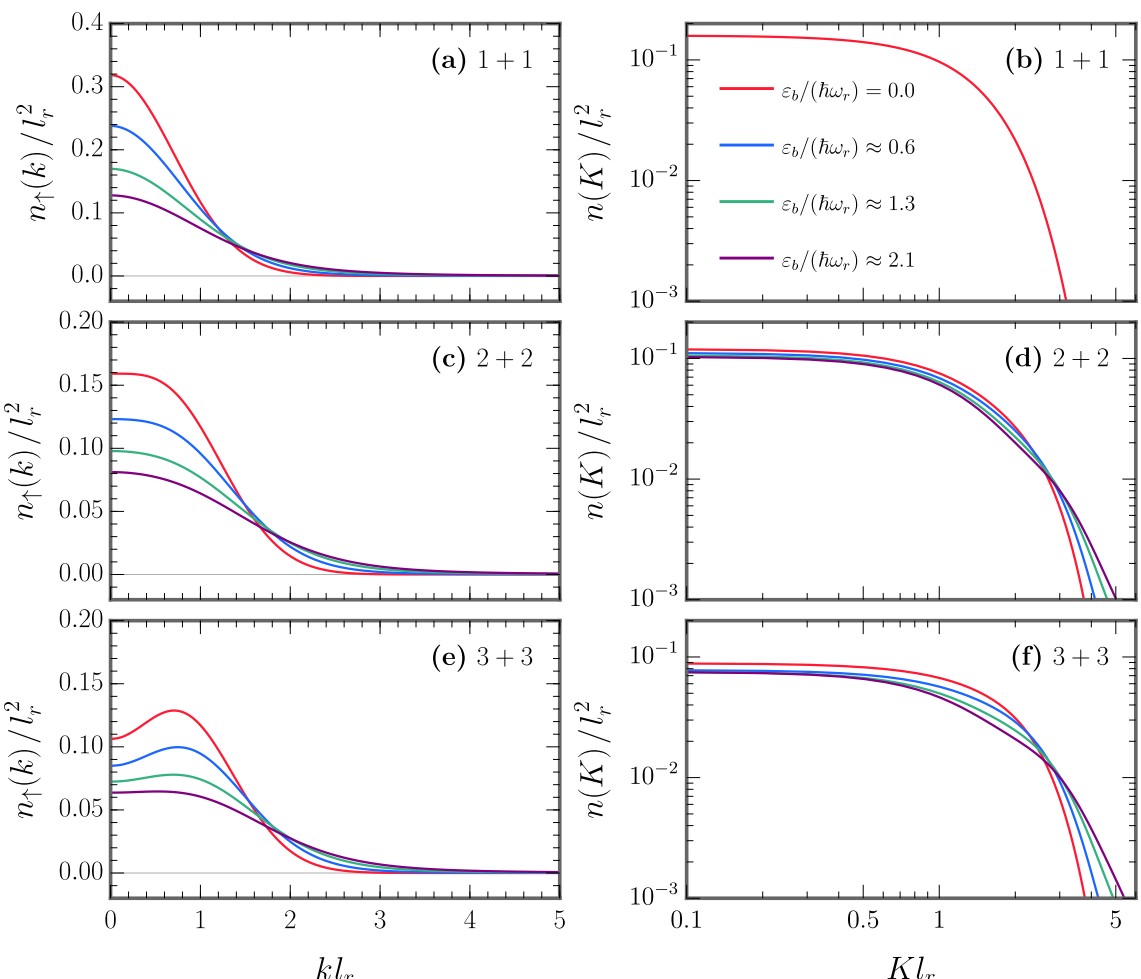

Figure 4: Left panels: The momentum distribution $n_\uparrow(k)$ (23) associated with the motion of spin-$\uparrow$ atoms for (a) $1+1$, (c) $2+2$, and (e) $3+3$ fermions in the ground state. Right panels: The momentum distribution $n(K)$ (32) associated with the centre-of-mass motion of spin-$\uparrow$-spin-$\downarrow$ pairs for (b) $1+1$, (d) $2+2$, and (f) $3+3$ fermions in the ground state [note the log-log scale]. The differently coloured lines correspond to different binding energies $\varepsilon_b$, while the effective range is fixed for all lines to very nearly zero, $r_{2D}/l_r^2 = -0.001 \approx 0$. By construction the two-body results for $1+1$ fermions in panel (b) are the same at all values of $\varepsilon_b$.

Analogously, the momentum distribution corresponding to the centre-of-mass motion of spin-$\uparrow$-spin-$\downarrow$ pairs is given by the Fourier transform of the reduced two-body density matrix defined in Eq. (16):

$$n(\mathbf{K}) = \frac{1}{(2\pi)^2} \int \int d\mathbf{R} \, d\mathbf{R}' \rho_{\mathrm{red}}(\mathbf{R}, \mathbf{R}') \exp\left[-i\mathbf{K}^T(\mathbf{R} - \mathbf{R}')\right]. \tag{32}$$

Here, we use the symbol $\mathbf{K}$ instead of $\mathbf{k}$ to distinguish the momentum vector associated with a pair from that of an atom. As with the calculation of the occupation numbers, the derivation of an analytical expression for $[n(K)]_{\mathbb{A}\mathbb{A}'}$ follows identically to the one above for $[n_\uparrow(k)]_{\mathbb{A}\mathbb{A}'}$ with a single minor adjustment. The transformation matrix $\mathbb{U}$ used to compute $\{c_1, c, c', a\}$ in Eq. (26) should be replaced by $\mathbb{U}'$ as explained in the text around Eqs. (19)–(20). Note that

the analysis in this subsection has been inspired by the corresponding three-dimensional calculation of Ref. [25] (see Appendix A therein).

In Fig. 4 we present the momentum distributions $[n_\uparrow(k)]_{\text{GS}}$ and $[n(K)]_{\text{GS}}$ for the ground state which were calculated by replacing $[\rho_\uparrow^m(r, r')]_{\mathbb{A}_i\mathbb{A}_j}$ with $[n_\uparrow(k)]_{\mathbb{A}_i\mathbb{A}_j}$ and $[n(K)]_{\mathbb{A}_i\mathbb{A}_j}$ in Eq. (13). In the non-interacting thermodynamic limit the momentum distribution for a single spin component features a 'step' at the Fermi momentum. However, when there are only very few atoms this step becomes 'smeared out' with a width determined by the radial harmonic trapping frequency $k_r \sim 1/l_r = \sqrt{m\omega_r/\hbar}$, as shown in panels (a), (c), and (e). Interestingly $n_\uparrow(k)$ adopts a distinct shape for each number of fermions, with the non-monotonicity in the $3+3$ case likely resulting from finite-size effects of the trap. By contrast, the distribution $n(K)$ displayed in panels (b), (d), and (f) varies little with either particle number or binding energy. For the particular case of $1+1$ fermions [Fig. 4(b)] $n(K)$ shows no dependence on the binding energy, mirroring the behaviour of the occupation numbers in Fig. 2(b). In three dimensions [25] $n(K)$ was found to exhibit two distinct features in the limit of small positive scattering length that could be associated with the condensation of pairs: a feature at smaller $K$ corresponding to the momentum distribution of non-interacting composite bosons of mass $2m$, and a feature at larger $K$ corresponding to the internal structure of the bosons. For our largest considered binding energy $\varepsilon_b \approx 2.1\hbar\omega_r$ we begin to see a 'shoulder' emerging at larger $K$ that resembles this phenomenon, however it is much less pronounced. This suggests — consistent with the previous subsections — that we remain far from the deep BEC regime.

## 3.4 Radial and Pair Distribution Functions

As well as density matrices, any *local* structural property $P(r)$ of the $N$-body system — such as a density profile or pair distribution function — can be calculated from the wave function as follows [25, 26, 28]:

$$P(r) = \int d\mathbf{r}' \frac{\delta(r - r')}{2\pi r'} \int d^{2N}\mathbf{x}\, \delta(\mathbf{r}' - \mathbf{x}) |\Psi(\mathbf{x})|^2. \tag{33}$$

Above, $\mathbf{r}'$ and $\mathbf{x}$ are 'dummy' integration variables used to perform the radial projection and many-body integration, with $\mathbf{x}$ representing a set of generalised co-ordinates such as the $N$ Jacobi position vectors described in Appendix A of Ref. [11]. The scalar $r$ denotes the radial distance associated with the chosen local observable. We define the averaged radial one-body density $P_\uparrow(r)$ by setting $r = |\mathbf{r}_1|$ in Eq. (33),[2] and also the averaged radial pair distribution function $P_{\uparrow\downarrow}(r)$ by setting $r = |\mathbf{r}_1 - \mathbf{r}_2|$. These quantities are normalised such that

$$2\pi \int_0^\infty dr\, r P_\uparrow(r) = 1 \quad \text{and} \quad 2\pi \int_0^\infty dr\, r P_{\uparrow\downarrow}(r) = 1. \tag{34}$$

The value of $2\pi r P_\uparrow(r)\, dr$ therefore equals the probability of locating a particle at a distance between $r$ and $r + dr$ from the centre of the trap. Likewise, the value of $2\pi r P_{\uparrow\downarrow}(r)\, dr$ equates to the probability of locating a spin-up particle and a spin-down particle at a distance between $r$ and $r + dr$ from each other.

We compute the ground-state matrix element $[P_\sigma(r)]_{\text{GS}}$ ($\sigma \equiv \uparrow$ or $\uparrow\downarrow$) in a similar manner to Eq. (13). In the explicitly correlated Gaussian basis, the matrix elements for arbitrary one- and two-body operators are respectively given by

$$\langle \phi_{\mathbb{A}_i} | V(\mathbf{r}_k) | \phi_{\mathbb{A}_j} \rangle = \mathbb{O}_{\mathbb{A}_i\mathbb{A}_j} \frac{b_k}{2\pi} \int d\mathbf{r}\, V(\mathbf{r}) \exp\left(-\frac{1}{2} b_k r^2\right), \tag{35a}$$

---

[2] Because the Fermi gases of interest are spin-balanced, the radial one-body densities for the spin-up and spin-down atoms are equal, $P_\uparrow(r) = P_\downarrow(r)$. In addition, since we consider only the sector of zero total orbital angular momentum, $P_\uparrow(r)$ is radially (circularly) symmetric.

$$\langle\phi_{\mathbb{A}_i}|V(\mathbf{r}_k-\mathbf{r}_l)|\phi_{\mathbb{A}_j}\rangle = \mathbb{O}_{\mathbb{A}_i\mathbb{A}_j}\frac{b_{kl}}{2\pi}\int d\mathbf{r}\,V(\mathbf{r})\exp\left(-\frac{1}{2}b_{kl}r^2\right), \tag{35b}$$

where

$$\frac{1}{b_k}=\left[\boldsymbol{\omega}^{(k)}\right]^T(\mathbb{A}_i+\mathbb{A}_j)^{-1}\boldsymbol{\omega}^{(k)}, \quad \left[\boldsymbol{\omega}^{(k)}\right]_p=(\mathbb{U}^{-1})_{kp}, \tag{36a}$$

$$\frac{1}{b_{kl}}=\left[\boldsymbol{\omega}^{(kl)}\right]^T(\mathbb{A}_i+\mathbb{A}_j)^{-1}\boldsymbol{\omega}^{(kl)}, \quad \left[\boldsymbol{\omega}^{(ij)}\right]_p=(\mathbb{U}^{-1})_{ip}-(\mathbb{U}^{-1})_{jp}, \tag{36b}$$

and $p=1,\ldots,N$ [11,14]. Correspondingly, we substitute $V(\mathbf{r}_k)=\delta(\mathbf{r}-\mathbf{r}_k)$ into Eq. (35a) to evaluate $[P_\uparrow(r)]_{\mathbb{A}_i\mathbb{A}_j}$ and $V(\mathbf{r}_k-\mathbf{r}_l)=\delta(\mathbf{r}-\mathbf{r}_k-\mathbf{r}_l)$ into Eq. (35b) to determine $[P_{\uparrow\downarrow}(r)]_{\mathbb{A}_i\mathbb{A}_j}$.

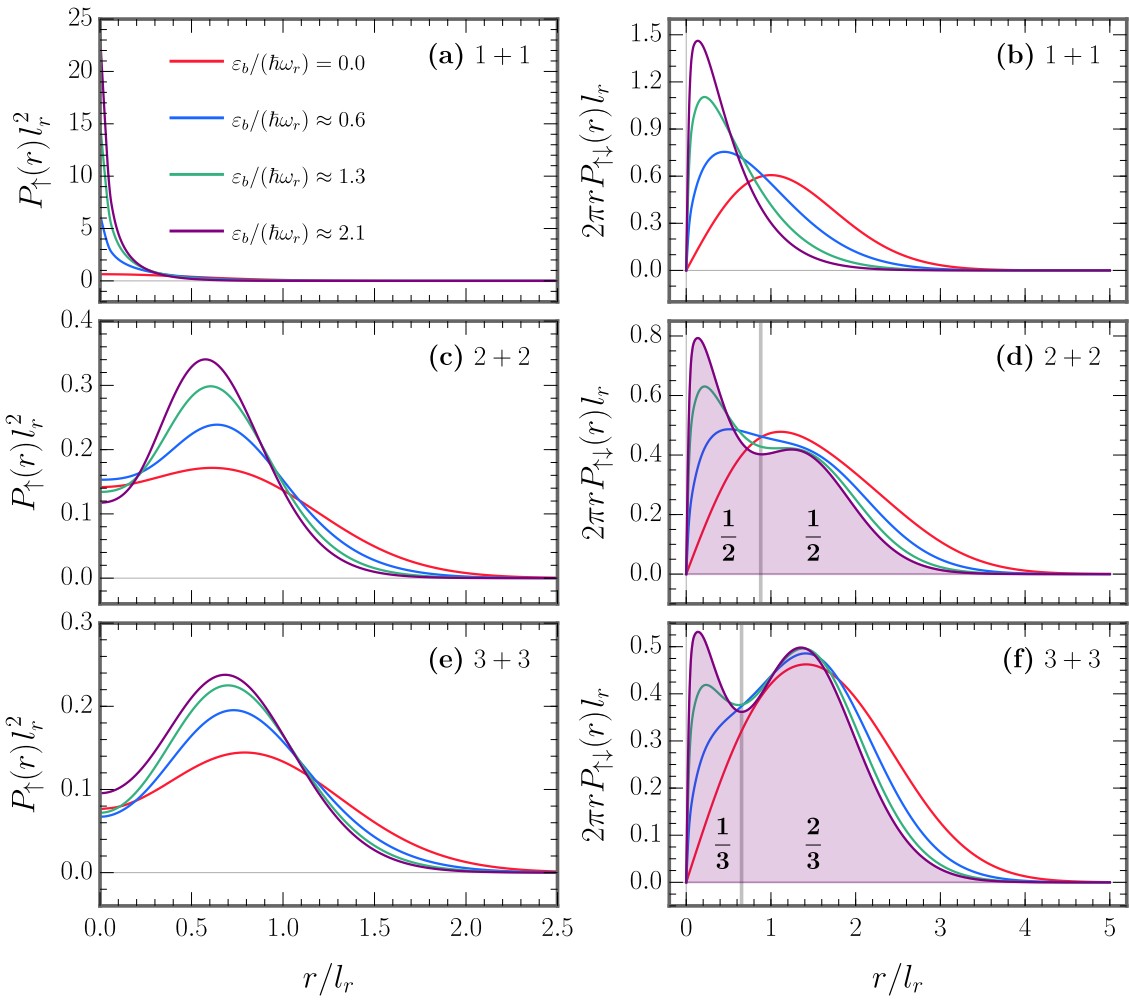

Figure 5: Left panels: The radial one-body density $P_\uparrow(r)$ for (a) $1+1$, (c) $2+2$, and (e) $3+3$ fermions in the ground state. Right panels: The (scaled) radial pair distribution function $P_{\uparrow\downarrow}(r)$ for (b) $1+1$, (d) $2+2$, and (f) $3+3$ fermions in the ground state. The results are shown for a variety of binding energies $\varepsilon_b$ at close to zero effective range, $r_{2D}/l_r^2 = -0.001 \approx 0$. The bold fractions indicate the (approximate) shaded area under the curve on either side of the grey vertical line for $\varepsilon_b \approx 2.1\hbar\omega_r$, as discussed in the main text.

Our results for the radial one-body density are shown in panels (a), (c), and (e) of Fig. 5. For the $1+1$ system at $\varepsilon_b = 0$, the spin-up atom occupies the non-interacting two-dimensional harmonic oscillator ground state, so $P_\uparrow(r)$ has a Gaussian radial profile with a maximum at the trap centre. For increasing $\varepsilon_b$, although the centre-of-mass motion of the pair remains governed by the external confinement, the attractive interactions confine the relative motion to shorter length scales, leading to deviations from a purely Gaussian profile and increasing the peak value of $P_\uparrow(r)$ at $r = 0$. On the linear scale used in Fig. 5(a), the comparatively small peak height of the $\varepsilon_b = 0$ profile causes its Gaussian decay to appear flattened relative to the interacting cases.

For the $2+2$ and $3+3$ systems the peak value of $P_\uparrow(r)$ shifts from the centre of the trap to a finite radius on the order of the radial harmonic trap length $l_r$, which sets the average interparticle spacing. This shift from zero to finite $r$ with increasing particle number reflects both the residual shell structure of the two-dimensional harmonic oscillator and the Pauli exclusion principle. The first harmonic oscillator shell is fully occupied for $N_\uparrow = 1$, whereas fermions occupy the first two shells for both $N_\uparrow = 2$ and $N_\uparrow = 3$, leading to similar behaviour in these cases: namely, $P_\uparrow(r)$ retains a single maximum that moves outward from the trap centre in order to accommodate both radial symmetry and Pauli repulsion between identical spins.

Panels (b), (d), and (f) of Fig. 5 show our results for the (scaled) radial pair distribution function. At binding energies of $\varepsilon_b \gtrsim \hbar\omega_r$ and when there is more than one particle per spin state, $rP_{\uparrow\downarrow}(r)$ develops a clear two-peak structure similar to what has been observed in three dimensions [25, 26]. The peak at smaller $r$ (around $0.1l_r$) signifies the formation of weakly bound dimers, while the peak at larger $r$ (between $1l_r$ and $2l_r$) is set by the dimer-dimer distance which is longer due to Pauli repulsion between same-spin fermions. The $2+2$ system has two such small interspecies distances (the distance between a spin-up and spin-down particle within a pair) and two large interspecies distances (the distance between a spin-up and spin-down particle in different pairs). Accordingly, if we integrate $P_{\uparrow\downarrow}(r)$ for $N_\uparrow = 2$ from zero up to the $r$ value where $rP_{\uparrow\downarrow}(r)$ features a minimum, then we find that the probability of forming a molecule (of being at short distances) is $\sim 1/2$ [26]. On the other hand, the $3+3$ system has three small interspecies distances and six large interspecies distances — and performing a similar integration confirms the probability of forming a molecule to be $\sim 1/3$. These probabilities are indicated in the figure. If we were to access the deep BEC regime $\varepsilon_b \gg 2\hbar\omega_r$, then the peak at smaller $r$ would become taller and narrower, while the peak at larger $r$ would become shorter and broader, with the pair density in between them reducing almost to zero — and the fractions mentioned above would become exactly $1/2$ and $1/3$ [26]. The reason why the scaled pair distribution function vanishes for $r \to 0$ is because we are using a finite-range interaction potential, such that unlike spins cannot approach each other at distances $\lesssim r_0$. If we had instead considered zero-range interactions, then the amplitude of $rP_{\uparrow\downarrow}(r)$ would have been finite at $r = 0$ [25, 44].

## 3.5 Finite Effective Range Effects

In this subsection we examine how the effective range influences the energetic and structural properties of the $3+3$ Fermi system. Figures 6–9 present results for a comparatively large negative effective range, $r_{2D}/l_r^2 = -0.2$ — corresponding to the most negative value considered in Ref. [11] — overlaid with our earlier results obtained for an almost vanishing effective range, $r_{2D}/l_r^2 = -0.001 \approx 0$.

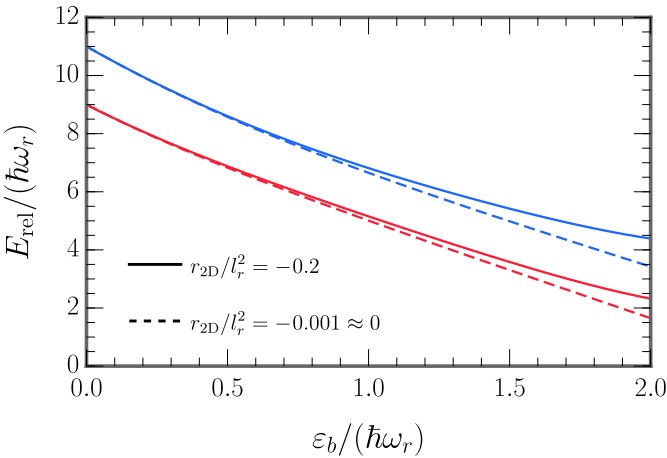

Figure 6: Energy of relative motion for the ground [red] and first excited state [blue] of the $3+3$ system as a function of the two-body binding energy in the monopole sector of zero total orbital angular momentum. Solid lines correspond to an effective range of $r_{2D}/l_r^2 = -0.2$ and dashed lines to $r_{2D}/l_r^2 = -0.001 \approx 0$.

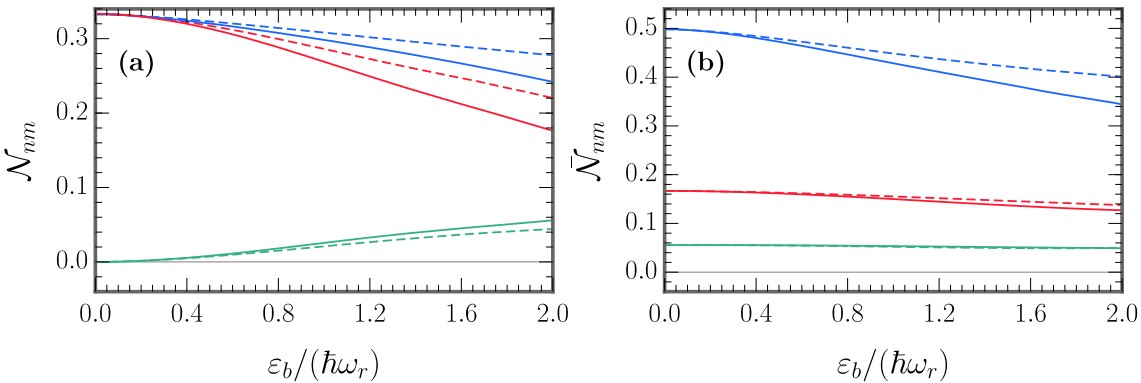

Figure 7: Occupation numbers of the one-body density matrix (a) and the reduced two-body density matrix (b) as functions of the binding energy for the $3+3$ fermion ground state. Results are shown for quantum numbers $n = 0$ and $|m| = 0$, 1, 2 (blue, red, and green curves, respectively). Solid lines correspond to an effective range of $r_{2D}/l_r^2 = -0.2$, while dashed lines correspond to $r_{2D}/l_r^2 = -0.001 \approx 0$, with the latter taken from the two lowest panels of Fig. 2.

The effective range $r_{2D}$ quantifies the leading energy-dependent correction in the low-energy description of two-dimensional scattering, entering the phase shift expansion via the term proportional to $k^2$ in Eq. (4). Consequently, at a given interaction strength (here parameterised by the binding energy $\varepsilon_b$) changes in $r_{2D}$ predominantly affect observables when the characteristic relative momenta in the $N$-body state are appreciable. As the binding energy is increased, this characteristic momentum scale grows because stronger pairing localises the relative two-body wave function in real space, which by Fourier duality necessarily broadens its distribution in momentum space. The influence of the effective-range term is therefore expected to become

more pronounced at higher $\varepsilon_b$. In other words, the nature of the $N$-body state and the strength of the binding energy set the characteristic relative momentum scale relevant for the two-body interaction in the system, while the effective range determines how strongly that interaction varies with energy, or equivalently with relative momentum, at that scale. This energy dependence arises in quasi-two-dimensional geometries because the finite axial extent of the wave function introduces an additional length scale $l_z$ into the collision process, causing the scattering amplitude to depend on the relative collision energy through the dimensionless combination $k l_z$. This connection is quantified by the mapping in Eq. (5), which relates $r_{2D}$ directly to the axial confinement length $l_z$ in the regime of strong axial confinement, $k l_z \ll 1$ [21–24].

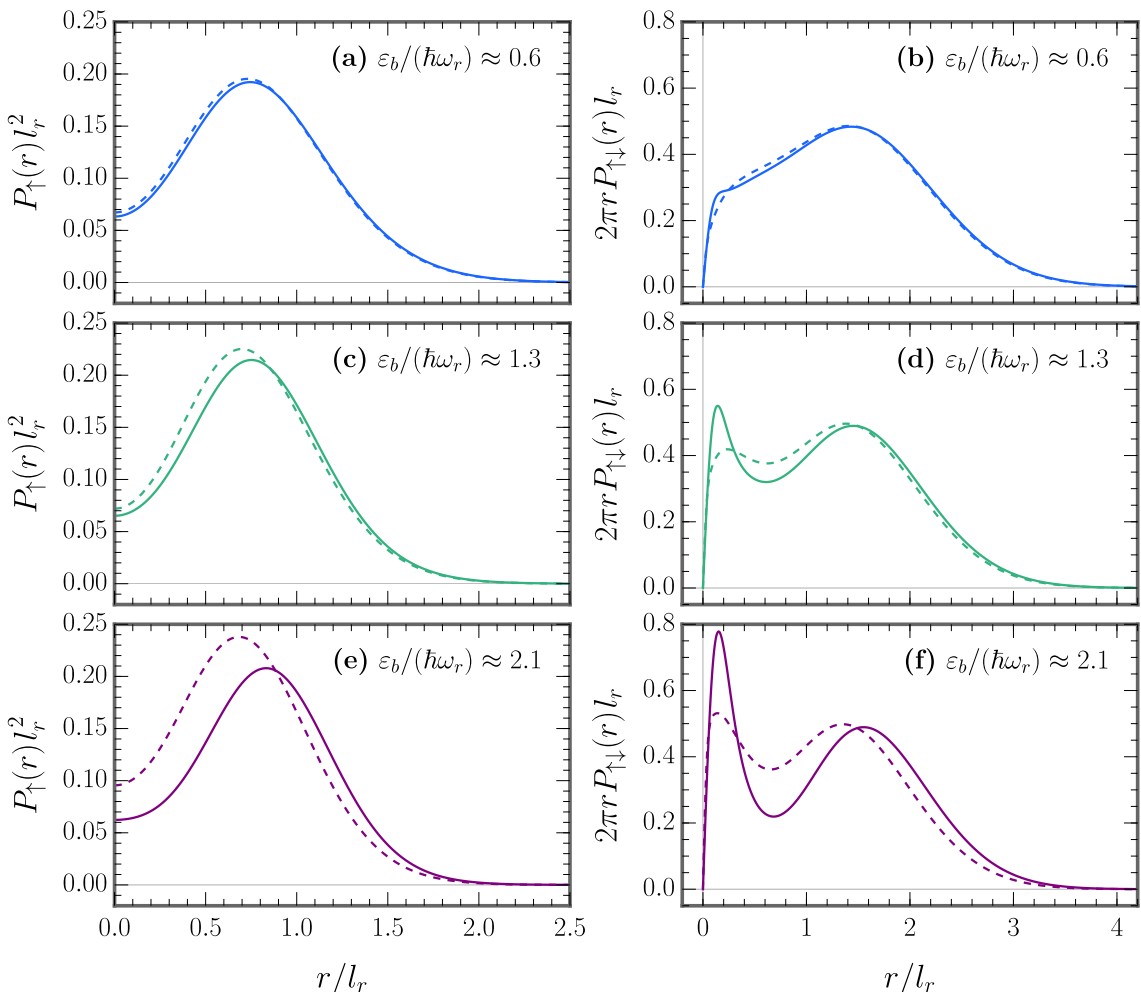

Figure 8: The radial one-body density [panels (a), (c), (e)] and the (scaled) radial pair distribution function [panels (b), (d), (f)] for the $3+3$ fermion ground state at three binding energies. Solid lines correspond to an effective range of $r_{2D}/l_r^2 = -0.2$ and dashed lines to $r_{2D}/l_r^2 = -0.001 \approx 0$, with the latter taken from the two lowest panels of Fig. 5.

Figure 6 shows the shifts in the low-lying energy levels of the $3+3$ system induced by a finite effective range $r_{2D}$ as a function of the binding energy $\varepsilon_b$. Consistent with the physical interpretation outlined above, these energetic shifts are smallest at low binding energy and be-

come larger as the binding energy increases, reflecting the growing importance of the $k^2 r_{2D}$ correction at higher characteristic relative momenta. At a given $\varepsilon_b$ these shifts are larger for the first excited state than for the ground state. The ground state has the smoothest real-space structure (i.e., the fewest spatial oscillations) compatible with the antisymmetry requirements and is therefore dominated by long length scales and a relatively narrow momentum distribution. By contrast, the first excited state must be orthogonal to the ground state and therefore exhibits additional spatial structure in the relevant relative co-ordinates, such as extra nodes and more rapid oscillations. These larger gradients in the wave function correspond to a broader momentum distribution with increased weight at higher relative momenta, rendering the excited state more sensitive to the $k^2 r_{2D}$ effective-range correction.

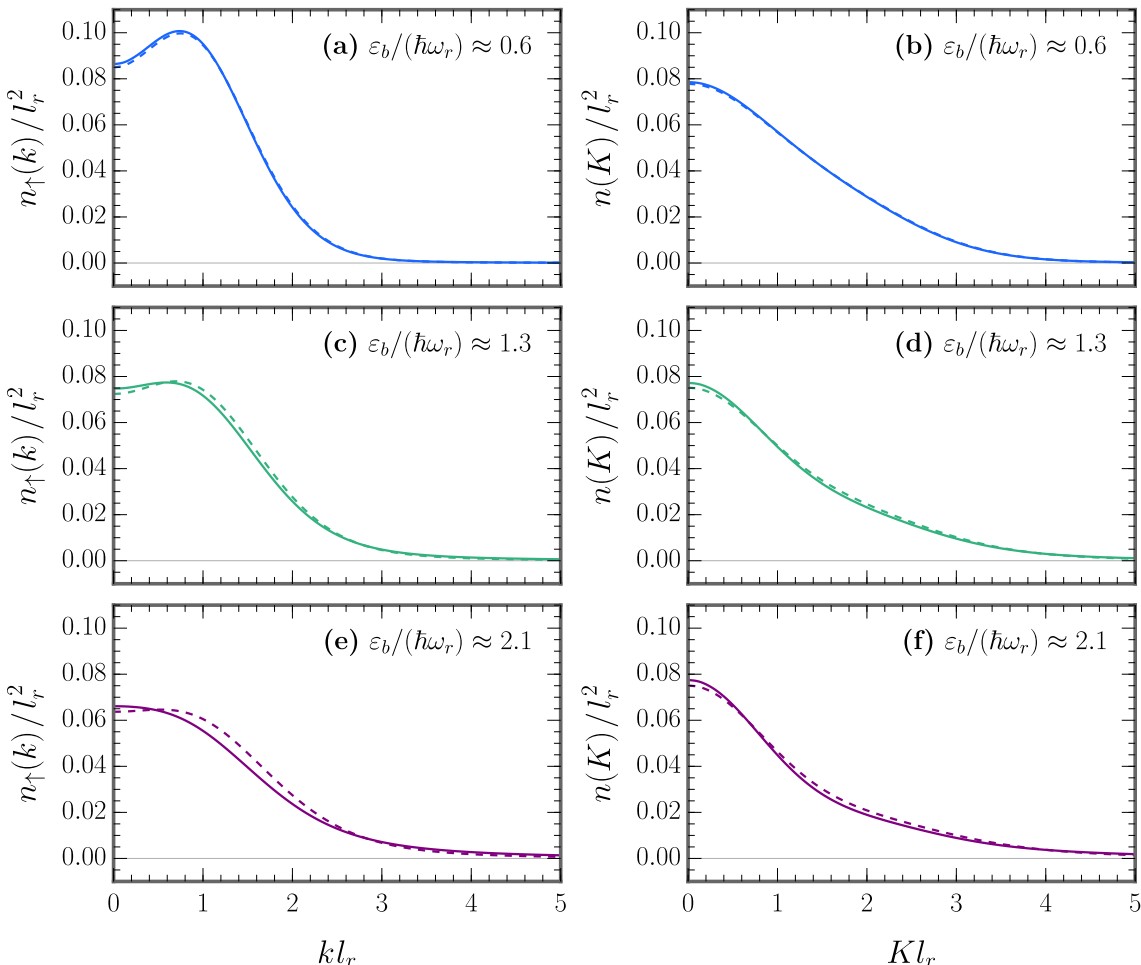

Figure 9: Ground-state momentum distribution for the motion of spin-↑ atoms [panels (a), (c), (e)] and the centre-of-mass motion of spin-↑-spin-↓ pairs [panels (b), (d), (f)] in the $3+3$ Fermi system at three binding energies. Solid lines correspond to an effective range of $r_{2D}/l_r^2 = -0.2$ and dashed lines to $r_{2D}/l_r^2 = -0.001 \approx 0$, with the latter taken from the two lowest panels of Fig. 4.

Structurally, a more negative effective range reduces the occupation of the lowest dominant natural orbitals of the one-body density matrix shown in Fig. 7, implying a redistribution

of spectral weight into higher excited orbitals. The representation of a tightly bound composite bosonic wave function using effective single-particle states (the natural orbitals) requires significantly more terms than the description of a weakly correlated antisymmetric fermionic wave function. This change in occupation therefore suggests that increasing the magnitude of the negative effective range drives the system closer to the BEC limit. The (scaled) radial pair distribution function displayed in Fig. 8 supports this interpretation: increasing $|r_{2D}|$ enhances the weight of the short-range peak at small $r$, indicating an increased probability of molecular pair formation. A complementary aspect of this behaviour appears in the one-body radial densities shown in the same figure. Tuning the effective range to more negative values at fixed binding energy shifts the density maximum outward and slightly reduces its height. In a harmonic trap the natural orbitals possess distinct radial structures and peak at different radii. Increasing $|r_{2D}|$ transfers a small amount of weight away from the most centrally peaked (lowest) orbitals and redistributes it among other higher orbitals with less central weight, so that the resulting superposition produces a density profile with an outward-shifted maximum. The effective-range dependence of these structural properties becomes more pronounced with increasing $\varepsilon_b$, reflecting the same underlying mechanism that governs the energetic shifts. Once the binding energy sets a characteristic relative momentum scale, a more negative effective range causes the interaction to act more strongly on the high-momentum components of the relative two-body wave function, thereby stabilising more compact molecule-like correlations and driving the system toward the BEC regime.

   Figure 9 shows that the single-particle and pair momentum distributions are only weakly affected by changes in the effective range. This is expected because the effective range enters as an energy-dependent correction that primarily modifies short-distance opposite-spin correlations, which are governed by high relative momenta. Although making $r_{2D}$ more negative increases the influence of the interaction on these high-momentum components, the associated spectral weight is confined to the tails of the momentum distributions. These tails contain very little probability compared with the low- and intermediate-momentum sectors that dominate the distributions, leaving the overall profiles essentially unchanged.

# 4   Conclusions

In this work we reported a numerically exact study of the energetic and structural properties of harmonically trapped, spin-balanced two-component Fermi gases in two dimensions, containing up to six particles. Using the explicitly correlated Gaussian method within a stochastic variational framework, we computed ground- and low-lying excited-state energy spectra, analysed non-local correlations through the one- and two-body density matrices, extracted atomic and pair momentum distributions, and characterised local structure via radial and pair distribution functions. We further examined how these observables are modified by a finite effective range, thereby connecting strictly two-dimensional and quasi-two-dimensional geometries relevant to current experiments.

   A limitation of our approach — which is shared by Ref. [25], a related investigation carried out in three dimensions — arises in the treatment of the reduced two-body density matrix. Owing to the large number of degrees of freedom involved, we restrict our analysis to correlations between spin-↑ and spin-↓ fermions evaluated at the same relative-distance vector. As a consequence, correlations that are non-local in the relative co-ordinate are neglected. This approximation affects the extracted occupation numbers $\bar{\mathcal{N}}_{nm}$, the pair momentum distribution $n(K)$, and the molecular condensate fraction $\mathcal{N}_{cond}$.

Across all observables, our results consistently reveal the emergence and gradual strengthening of pair correlations as the two-body binding energy is increased. Despite this clear enhancement of pairing, we do not observe the onset of a deep BEC regime in which the system crosses over to a description in terms of tightly bound composite bosons. As discussed in detail in Sections 2, 3.1, and 3.2.3, this absence has both physical and methodological origins. Physically, in a trapped few-fermion system the notion of a many-body BEC limit is inherently ill-defined: the small particle number, large density fluctuations, and discrete single-particle spectrum prevent the clear separation of length and energy scales that characterises a genuine many-body condensate. Methodologically, extending the calculations to larger binding energies would require resolving increasingly tightly bound $\uparrow\downarrow$ pairs whose internal length scales become much smaller than the trap length. This separation of length scales necessitates basis functions with both very small and very large spatial extents, leading to a rapid growth in the size of the explicitly correlated Gaussian basis required for convergence. This computational burden is compounded by the stochastic nature of the basis-optimisation process, since many trial basis functions must be generated and discarded before converged energies are obtained. Thus, our results emphasise an intermediate binding-energy regime in which fully converged calculations are feasible for all particle numbers considered.

In addition to the binding energy, the range of accessible particle numbers is limited by numerical considerations. At any interaction strength the dominant contribution to the runtime arises from enforcing the antisymmetry of the $N$-body wave function, which involves summing over all permutations of identical fermions. The number of such permutations grows factorially with particle number, leading to a rapid increase in computational time [11]. As a result, evaluating matrix elements for $4+4$ or more fermions becomes prohibitively time-consuming, and even computing the first several excited states for $3+3$ fermions at $\varepsilon_b \lesssim 2\hbar\omega_r$ requires very long runtimes. This implies that the explicitly correlated Gaussian method would not be well suited to performing calculations at finite temperature (where the density matrix becomes a sum over the ground and excited states, weighted by the Boltzmann factor) or to performing time dynamics (where the original wave function is projected onto a new time-evolved basis, potentially acquiring non-zero excited-state components).

# Acknowledgements

The authors would like to thank Xiangyu (Desmond) Yin for writing the initial version of the C code used to diagonalise the Hamiltonian within the explicitly correlated Gaussian framework.

**Funding Information**   This research was supported by the Australian Research Council Centre of Excellence in Future Low-Energy Electronics and Technologies, also known as 'FLEET' (Project No. CE170100039). Emma Laird received funding from a Women−in−FLEET research fellowship.

# A   Analytical Results in the Non-Interacting Limit

In this appendix we analytically derive all the occupation numbers of the projected one-body density matrix $\mathcal{N}_{nm}$ and the projected reduced two-body density matrix $\bar{\mathcal{N}}_{nm}$ for the trapped, non-interacting $2+2$ atomic Fermi gas in the ground state. In two-dimensional position space

the ground-state wave function is

$$\Psi_{2+2}^{(GS)}(\mathbf{r}_1^\uparrow, \mathbf{r}_2^\downarrow, \mathbf{r}_3^\uparrow, \mathbf{r}_4^\downarrow) = \frac{1}{\sqrt{2}\pi^2 l_r^6} \exp\left[-\sum_{i=1}^4 \frac{(\mathbf{r}_i^\sigma)^2}{2l_r^2}\right] (\mathbf{r}_1^\uparrow - \mathbf{r}_3^\uparrow)^T (\mathbf{r}_2^\downarrow - \mathbf{r}_4^\downarrow), \qquad (A.1)$$

with $\sigma = \uparrow, \downarrow$. It can readily be confirmed that Eq. (A.1) is normalised and correctly gives a total ground-state energy of $E_{com} + E_{rel} = 6\hbar\omega_r$. As defined in Eq. (6) the corresponding one-body density matrix is

$$[\rho_\uparrow(\mathbf{r}, \mathbf{r}')]_{GS} = \int \cdots \int d\mathbf{r}_2^\downarrow d\mathbf{r}_3^\uparrow d\mathbf{r}_4^\downarrow \, \Psi_{2+2}^{(GS)}(\mathbf{r}, \mathbf{r}_2^\downarrow, \mathbf{r}_3^\uparrow, \mathbf{r}_4^\downarrow) \, \Psi_{2+2}^{(GS)*}(\mathbf{r}', \mathbf{r}_2^\downarrow, \mathbf{r}_3^\uparrow, \mathbf{r}_4^\downarrow)$$

$$= \frac{1}{2\pi} \exp\left\{-\frac{1}{2}\left[\mathbf{r}^2 + (\mathbf{r}')^2\right]\right\}(1 + \mathbf{r}^T \mathbf{r}'). \qquad (A.2)$$

Writing $\mathbf{r}^T \mathbf{r}' = rr'\cos(\theta - \theta')$ and then applying Eq. (11) yields

$$[\rho_\uparrow^{m=0}(r, r')]_{GS} = \exp\left\{-\frac{1}{2}\left[r^2 + (r')^2\right]\right\}, \qquad (A.3a)$$

$$[\rho_\uparrow^{m=\pm 1}(r, r')]_{GS} = \frac{1}{2} \exp\left\{-\frac{1}{2}\left[r^2 + (r')^2\right]\right\} rr', \qquad (A.3b)$$

$$[\rho_\uparrow^{m \geq 2}(r, r')]_{GS} = 0. \qquad (A.3c)$$

Finding the eigenvalues of $\sqrt{r}\,[\rho_\uparrow^m(r, r')]_{GS}\sqrt{r'}\Delta r$ affords $\mathcal{N}_{0,0} = 1/2$ and $\mathcal{N}_{0,\pm 1} = 1/4$ (with all other occupation numbers zero), consistent with the left middle panel of Fig. 2.

Similarly, the relevant two-body density matrix is

$$[\rho(\mathbf{r}_\uparrow, \mathbf{r}'_\uparrow; \mathbf{r}_\downarrow, \mathbf{r}'_\downarrow)]_{GS} = \int \cdots \int d\mathbf{r}_3^\uparrow d\mathbf{r}_4^\downarrow \, \Psi_{2+2}^{(GS)}(\mathbf{r}_\uparrow, \mathbf{r}_\downarrow, \mathbf{r}_3^\uparrow, \mathbf{r}_4^\downarrow) \, \Psi_{2+2}^{(GS)*}(\mathbf{r}'_\uparrow, \mathbf{r}'_\downarrow, \mathbf{r}_3^\uparrow, \mathbf{r}_4^\downarrow)$$

$$= \frac{1}{4\pi^2} \exp\left\{-\frac{1}{2}\left[\mathbf{r}_\uparrow^2 + (\mathbf{r}'_\uparrow)^2 + \mathbf{r}_\downarrow^2 + (\mathbf{r}'_\downarrow)^2\right]\right\}\left\{1 + \mathbf{r}_\uparrow^T \mathbf{r}'_\uparrow + \mathbf{r}_\downarrow^T \mathbf{r}'_\downarrow + 2(\mathbf{r}_\uparrow^T \mathbf{r}_\downarrow)\left[(\mathbf{r}'_\uparrow)^T \mathbf{r}'_\downarrow\right]\right\}, \qquad (A.4)$$

as defined in Eq. (15). By transforming to the centre-of-mass and relative co-ordinates of the two spin-$\uparrow$-spin-$\downarrow$ pairs, we arrive at

$$[\rho(\mathbf{R}, \mathbf{R}'; \mathbf{r}, \mathbf{r}')]_{GS} = \frac{1}{32\pi^2} \exp\left\{-\left[\mathbf{R}^2 + (\mathbf{R}')^2\right] - \frac{1}{4}\left[\mathbf{r}^2 + (\mathbf{r}')^2\right]\right\} \times$$

$$\left\{8 + 16\mathbf{R}^T \mathbf{R}' + 4\mathbf{r}^T \mathbf{r}' + \left(4\mathbf{R}^2 - \mathbf{r}^2\right)\left[4(\mathbf{R}')^2 - (\mathbf{r}')^2\right]\right\}. \qquad (A.5)$$

Setting $\mathbf{r} = \mathbf{r}'$ and subsequently integrating over $\mathbf{r}$ leads to

$$[\rho_{red}(\mathbf{R}, \mathbf{R}')]_{GS} = \frac{1}{2\pi} \exp\left\{-\left[\mathbf{R}^2 + (\mathbf{R}')^2\right]\right\}\left[3 + 2\mathbf{R}^2(\mathbf{R}')^2 - (\mathbf{R} - \mathbf{R}')^T(\mathbf{R} - \mathbf{R}')\right]. \qquad (A.6)$$

At this point, we can expand $(\mathbf{R} - \mathbf{R}')^T(\mathbf{R} - \mathbf{R}') = R^2 + (R')^2 - 2RR'\cos(\phi - \phi')$ and perform partial-wave projections in analogy to Eq. (11) to find that

$$[\rho_{red}^{m=0}(R, R')]_{GS} = \exp\left\{-\left[R^2 + (R')^2\right]\right\}\left\{3 + 2(RR')^2 - \left[R^2 + (R')^2\right]\right\}, \qquad (A.7a)$$

$$[\rho_{red}^{m=\pm 1}(R, R')]_{GS} = \exp\left\{-\left[R^2 + (R')^2\right]\right\}RR', \qquad (A.7b)$$

$$[\rho_{red}^{m \geq 2}(R, R')]_{GS} = 0, \qquad (A.7c)$$

where $\phi^{(')}$ is the angle associated with the vector $\mathbf{R}^{(')}$. The occupation numbers can now be obtained as the eigenvalues of $\sqrt{R}\,[\rho_{red}^m(R, R')]_{GS}\sqrt{R'}\Delta R$. The first of the above relations (A.7a) gives $\bar{\mathcal{N}}_{0,0} = 0.625$ and $\bar{\mathcal{N}}_{1,0} = 0.125$, and the second (A.7b) gives $\bar{\mathcal{N}}_{0,\pm 1} = 0.125$, while all other occupation numbers vanish — in agreement with the right middle panel of Fig. 2.

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
