# Peer review of "Energetic and Structural Properties of Two-Dimensional Trapped Mesoscopic Fermi Gases"

_SciPost Physics Core_

## Round 1 · Referee Report · Anonymous (Referee 1) · 2025-8-22

Strengths

1- the scope of the paper is broad for an exact few body calculation. Covering both energetic properties and a range of correlation measures. 2-The paper has a clear relevance to ongoing experimental work. 3-Novel insights into the finite effective range effects.

Weaknesses

1- The condenstate fraction is adapted from the 3D few -fermion definition to the 2D few-body case. In your results the behaviour is non-monotonic and shows only a small variation with binding energy, unlike the monotonic growth seen in 3D. It was unclear to me why this is the case.

Report

Overall, I found the paper to be a strong and valuable contribution. The presentation is clear and the connections to current experiments is a strength.

Requested changes

1-Could the authors explain more clearly why the 3D definition does not work as well in 2D few-body systems. As this was unclear to me.

Recommendation

Publish (easily meets expectations and criteria for this Journal; among top 50%)

  • validity: high
  • significance: top
  • originality: high
  • clarity: top
  • formatting: perfect
  • grammar: excellent

Author:  Emma Laird  on 2025-12-25  [id 6186]

(in reply to Report 1 on 2025-08-22)

We thank the referee for their careful reading of our paper and for their encouraging feedback. For clarity, all changes made in response to both referee reports are highlighted in orange in the revised manuscript attached to this comment. We also apologise for the delay in resubmission, which was due to unforeseen circumstances on our side.

Regarding your requested change:

We have completely rewritten and expanded Section 3.2.3 on the ‘Molecular Condensate Fraction’. In its revised form, this Section:
i) provides a clearer rationale for using the chosen definition of the condensate fraction in two dimensions;
ii) explains why the measure remains meaningful after eliminating degrees of freedom from the reduced two-body density matrix;
iii) describes how the condensate fraction is expected to behave as a function of the two-body binding energy;
iv) offers an expanded interpretation of the numerical results in Fig. 3, including an explanation of the weak non-monotonicity at small binding energies.
[lines 332–398]

Our definition of the condensate fraction in two dimensions is based on the eigenvalues of the reduced two-body density matrix and is directly analogous to the three-dimensional formulation of Ref. [25] (Blume and Daily, 2011). Within this framework, the condensate fraction depends on the relative dominance of the leading eigenvalue over the rest of the spectrum, rather than on the absolute magnitude of that eigenvalue. Consequently, the condensate fraction is not strictly constrained to increase monotonically and may exhibit non-monotonic behaviour over a limited interaction range, despite an overall tendency to increase as pairing correlations strengthen.

The interaction window accessible in our calculations lies on the non-molecular side of the two-dimensional crossover, where we find that the condensate fraction is effectively flat for the 2+2 and 3+3 systems. In three dimensions the analogous regime corresponds to negative inverse scattering length, and Fig. 11 of Ref. [25] shows that the condensate fraction there is likewise essentially constant — so the 2D and 3D results are consistent.

We do not access the very strong interaction regime in two dimensions where the condensate fraction is expected to approach unity monotonically. As we now explain, reaching this regime would require resolving increasingly short-range structure in the pair wave function, leading to a rapid growth of the Gaussian basis size needed for convergence. Instead, our results focus on an intermediate range of binding energies for which fully converged calculations are attainable across all particle numbers considered.

Attachment:

2D_Few-Fermion_Systems_2D_Ref_1.pdf

---

## Round 1 · Referee Report · Anonymous (Referee 2) · 2025-9-20

Strengths

1 Few-fermion unpolarized systems are calculated.

2 Energetic and structural properties are found and reported in detail. Energy spectrum and occupation numbers are reported.

Weaknesses

1 It is not cleear why BEC regime is not fully reached. 2 Discussion of the effective-range effects can be improved

Report

Authors study properties of unpolarized few fermion system under harmonic confinement in two dimesions. Energetic and structural properties are described in detail. Effects of finite range are considered. The studied system is related to recent experiments with ultracold atoms.

While overall, the Manuscript is of good quality, still a number of points should be improved before it can be accepted.

Requested changes

Line 89, a very specific shape for the interaction potential is used, additional motivation or justification for this shape should be mentioned here

Line 112, 115, it might be appropriate to mention that the used definition of the scattering length implies logarithmic dependence of the wave function as ln(a_{2D} / r)

Fig 1, 6, “relative energy” E / (hbar omega) might sound ambiguous, I would suggest using the "energy of relative motion" instead to make it clear.

Line 220, “parameters … are optimized stochastically”, it is appropriate to specify which criteria for optimization are actually used

Line 230 Natural orbital analysis reminds me of this article Phys. Rev. A 68, 033602, 2003 which is not cited, as all references are limited to Ref. [25]. I think the Authors could add more references for the decomposition of the OBDM

Line 259 “these finite occupation numbers decrease”, it would be better to say explicitly what is meant by “these”

Fig 3, it is not clear if Ncond is expected to increase or decrease as the binding energy is increased

Line 381 “any local property …”, at this point I can imagine a potential energy or density profile, as an example of a local observable. Line 386, {\bf r}, is used but not defined in Eq. (33). I think the presentation can be improved. It is clear that a local observable can be written as an integral over the square of the wave function, but the knowledge of the observable should be introduced to Eq. (33)

Fig 5a, it is not clear why for zero Eb the density profile is different from a Gaussian.

Section 3.5 contains only 15 lines, while referring to four pages of figures. It feels like more discussion is needed, especially as Eq (5) is explicitly introduced to provide the value of the effective range. Is it possible to describe the seen effects in terms of r2D?

As a general observation, it is not clear why a deep BEC regime is not seen in this article.

Recommendation

Ask for minor revision

  • validity: high
  • significance: good
  • originality: good
  • clarity: high
  • formatting: excellent
  • grammar: perfect

Author:  Emma Laird  on 2025-12-25  [id 6187]

(in reply to Report 2 on 2025-09-20)

We thank the referee for their careful reading of our paper and for their constructive feedback. Below we respond to each of the points raised, which we have numbered in the order they appear in the referee report (from top to bottom). For clarity, all changes made in response to both referee reports are highlighted in orange in the revised manuscript attached to this comment. We also apologise for the delay in resubmission, which was due to unforeseen circumstances on our side.

1) In the revised manuscript we have added a paragraph that further clarifies and motivates our choice of interaction potential. This new text summarises the practical reasons for employing the two-Gaussian functional form and notes that, due to universality, its precise short-range shape is unimportant; it also cites earlier works which support that point. Rather than placing this explanation adjacent to Eq. (3), we have appended it to the end of the ‘Model’ Section where it naturally draws together several considerations introduced throughout that Section. [lines 148–159]

2) We have added a sentence below Eq. (4) to make it clear that our definition of the 2D scattering length leads to the expected logarithmic behaviour of the zero-energy relative radial wave function at large distances. [lines 105–107]

3) We have revised the captions of Figs. 1 and 6, ensuring that the term ‘relative energy’ is replaced with ‘energy of relative motion’ to avoid ambiguity. [pages 6, 19]

4) We have added two sentences to the paragraph immediately below Eq. (8) specifying the optimisation criterion used in the stochastic variational method (namely, energy minimisation). We also explain that the allowed ranges for stochastic variations of the Gaussian widths are chosen to reflect physically relevant interparticle length scales. [lines 235–239]

5) We have substantially expanded the discussion of the natural-orbital decomposition of the one-body density matrix. Immediately below Eq. (10) we have added a paragraph which outlines the historical development of the theory (Löwdin; Penrose and Onsager; Yang) and gives representative many-body and few-body cold-atom applications (DuBois and Glyde; Zöllner et al.). We have also supplemented the citation of Blume and Daily with a sentence noting that their work provides the corresponding 3D few-body fermionic formulation. These additions supply the broader context and address the referee’s concern regarding the appropriate citation of prior literature. [lines 247–255, 259–261]

6) We thank the referee for pointing out the ambiguity in the phrase ‘these finite occupation numbers’. We have revised the first sentence of the paragraph to explicitly identify which occupation numbers are meant. [lines 289–291]

7) We have completely rewritten and expanded Section 3.2.3 on the ‘Molecular Condensate Fraction’. In its revised form, this Section: i) provides a clearer rationale for using the chosen definition of the condensate fraction in two dimensions; ii) explains why the measure remains meaningful after eliminating degrees of freedom from the reduced two-body density matrix; iii) describes how the condensate fraction is expected to behave as a function of the two-body binding energy (addressing the referee’s exact concern); iv) offers an expanded interpretation of the numerical results in Fig. 3, including an explanation of the weak non-monotonicity at small binding energies. [lines 332–398]

8) We have revised the presentation around Eq. (33) to clarify the meaning of the ‘local observable’ and the associated variables. In particular, we now give explicit examples of local observables, define the role of the radial variable, and explain the interpretation of the integration variables appearing in the expression. [lines 442–450]

9) We have expanded the discussion of the one-body radial densities in Fig. 5. We now explicitly clarify that the result for the 1+1 system at zero binding energy is a Gaussian. We also explain that increasing the interaction strength increasingly localises the particles, causing the peak density at the trap centre to grow significantly. Consequently, on the linear scale used in Fig. 5(a) the zero-binding-energy profile appears flattened relative to the interacting cases. [lines 461–469]

10) We have completely rewritten and expanded Section 3.5 to provide a detailed physical interpretation of the effective-range effects shown in Figs. 6–9. The revised Section now explicitly connects the observed energetic and structural changes to the momentum dependence of the two-body interaction arising from the finite effective range (or, equivalently, from the finite axial extent of the wave function). In particular, we explain how increasing the magnitude of the negative effective range leads to systematic shifts in the energy spectra, a redistribution of natural-orbital occupations, and enhanced short-range pair correlations. We also clarify why the momentum distributions remain largely unchanged, despite these other observable effects. [lines 499–564]

11) In the revised manuscript we now emphasise more clearly that the absence of a deep BEC regime in our calculations has both physical and methodological origins. The physical origin — namely, the fact that a strongly interacting BEC limit is difficult to define in a trapped few-fermion system — is discussed in Section 2 [lines 117–126], while the methodological and computational limitations are addressed in Sections 3.1 [lines 202–218] and 3.2.3 [lines 384–398] where their impact is most clearly visible. These considerations are also reiterated and synthesised in a substantially revised Conclusion [lines 565–610]. (Note that the discussions in Sections 2 and 3.1 were already present in the originally submitted manuscript and are not new additions.)

Attachment:

2D_Few-Fermion_Systems_2D_Ref_2.pdf

---

## Round 2 · Author Response

Please find our revised manuscript attached. Responses to the referees are provided below each report, with all corresponding revisions highlighted in orange in the manuscript. Below, we list the line numbers where changes have been made; expanded explanations are given in our replies to the referees.
Thank you for your consideration.
Kind regards,
Emma

---

## Round 2 · List of Changes

- motivated the choice of interaction potential, lines 148–159
- clarified the logarithmic asymptotic behaviour of the wave function, lines 105–107
- resolved ambiguity in figure captions 1 and 6 on pages 6 and 19 (relative energy)
- specified the optimisation criterion in the stochastic variational method, lines 235–239
- expanded the contextual discussion of the natural-orbital decomposition, lines 247–255 and 259–261
- clarified the reference to finite occupation numbers, lines 289–291
- substantially revised the discussion of the molecular condensate fraction, lines 332–398
- clarified the definition of local observables, lines 442–450
- expanded the explanation of the Gaussian one-body radial density profiles, lines 461–469
- expanded the physical interpretation of the finite effective-range effects, lines 499–564
- substantially revised the conclusion, lines 565–610

---

## Editorial Decision

unknown